# Stalled translation by mitochondrial stress upregulates a CNOT4-ZNF598 ribosomal quality control pathway important for tissue homeostasis

Ji Geng [1,4] ✉, Shuangxi Li[1,2,4], Yu Li[1,2], Zhihao Wu[1], Sunil Bhurtel[1], Suman Rimal[1], Danish Khan [3], Rani Ohja[1], Onn Brandman [3] & Bingwei Lu [1] ✉

Translational control exerts immediate effect on the composition, abundance, and integrity of the proteome. Ribosome-associated quality control (RQC) handles ribosomes stalled at the elongation and termination steps of translation, with ZNF598 in mammals and Hel2 in yeast serving as key sensors of translation stalling and coordinators of downstream resolution of collided ribosomes, termination of stalled translation, and removal of faulty translation products. The physiological regulation of RQC in general and ZNF598 in particular in multicellular settings is underexplored. Here we show that ZNF598 undergoes regulatory K63-linked ubiquitination in a CNOT4-dependent manner and is upregulated upon mitochondrial stresses in mammalian cells and *Drosophila*. ZNF598 promotes resolution of stalled ribosomes and protects against mitochondrial stress in a ubiquitination-dependent fashion. In *Drosophila* models of neurodegenerative diseases and patient cells, ZNF598 overexpression aborts stalled translation of mitochondrial outer membrane-associated mRNAs, removes faulty translation products causal of disease, and improves mitochondrial and tissue health. These results shed lights on the regulation of ZNF598 and its functional role in mitochondrial and tissue homeostasis.

Cells respond to stress stimuli by reconfiguring their transcriptional, translational, and metabolic profiles. Transcriptomics and proteomics studies have revealed that transcript levels and protein abundance do not always match[1], emphasizing the importance of post-transcriptional control of protein output. Compared to transcriptional control, translational control of available mRNAs exerts an immediate effect on the composition, abundance, and integrity of the proteome[2], making it particularly important under stress conditions[3]. Moreover, mRNA translation is intrinsically a very energy-demanding process, and its regulation is intimately linked to the bioenergetic and metabolic status[3], making translational control essential for cellular homeostasis. Translational control critically influences cellular proliferation, growth, and survival, and impacts diverse physiological processes, from early development to synaptic plasticity[4,5]. Deregulated translational control is profoundly implicated in human diseases[6–11].

Although translation is known to be tightly controlled at the rate-limiting initiation step[12], the elongation and termination steps are also subject to intricate regulation[13,14]. During translation elongation, ribosome slowdown and stalling can occur for various reasons. Some are functional and serve to facilitate cellular dynamics, such as

[1]Department of Pathology, Stanford University School of Medicine, Stanford, CA 94305, USA. [2]Shandong Provincial Key Laboratory of Animal Cell and Developmental Biology, School of Life Sciences, Shandong University, Qingdao 266237, China. [3]Department of Biochemistry, Stanford University School of Medicine, Stanford, CA 94305, USA. [4]These authors contributed equally: Ji Geng, Shuangxi Li. ✉e-mail: jigeng@stanford.edu; bingwei@stanford.edu

co-translational protein folding, frameshifting, and subcellular protein targeting. Others are detrimental and can be triggered by damaged mRNAs, mRNA secondary structures, an insufficient supply of aminoacyl-tRNAs, or environmental stress[15,16]. Ribosome slowdown and stalling can result in ribosome collision[17], which is sensed by the cell as a proxy for aberrant translation and triggers ribosome-associated quality control (RQC)[18–22]. Key factors involved in the process are the ubiquitin ligase ZNF598 (Hel2 in yeast) and the 40S subunit protein Rack1 (Asc1 in yeast), which recognize the distinct 40S–40S interface of collided ribosomes and promote ubiquitination of specific 40S proteins[23,24], and the ASC complex that disassembles the leading collided ribosome[25,26]. This then triggers a series of downstream quality control events, including ribosome subunit splitting and recycling by ABCE1[27], CAT-tailing modification by NEMF (Tae2 or Rqc2 in yeast) of nascent peptide chains (NPCs) still attached to the 60S subunit[28], release of stalled NPCs from the peptidyl-tRNA/60S complex by ANKZF1 (Vms1 in yeast)[29,30], and clearance of stalled NPCs by the Ltn1 E3 ligase-mediated ubiquitination and proteasomal degradation[31,32].

Recent studies have highlighted the importance of regulatory ribosomal protein ubiquitination in the RQC process. RQC is initiated by ZNF598 through site-specific mono-ubiquitination of RPS10/eS10 and RPS20/uS10 of ribosomes stalled on aberrant mRNAs[23,24,33,34], leading to the proposal that the unique 40S–40S interface of collided ribosomes is specifically recognized by ZNF598 such that problematic translation is differentiated from normal translation. Collided ribosomes marked by ZNF598-dependent ubiquitination of RPS10 are disassembled by the conserved ASC-1 complex (ASCC) containing ASCC3, an ATP-dependent helicase[25,26]. In addition to RPS10 and Rps20, other 40S subunits such as Rps2 and Rps3 are also mono-ubiquitinated, with ZNF598 and RNF10 having been implicated in these events[23,24,35]. Moreover, at least in yeast, the small ribosomal protein eS7 is ubiquitinated by the E3 ligase CNOT4[36]. CNOT4 was originally identified as a component of the CCR4-NOT complex involved in transcription and RNA degradation[37], but could also act as a ribosome-associated E3 ligase that works together with another ribosome-associated E3 ligase Hel2 to control RQC-uncoupled no go decay outside the disome[38], and CNOT4 can modify the RQC related translation termination and ribosome-recycling factor ABCE1[39]. However, the role of CNOT4 in handling stalled translation remains poorly understood. Regulatory ribosome ubiquitination involved in RQC is also modulated at the deubiquitination step, with several enzymes having been implicated, including USP10[40], OTUD3[41], and USP21[41] that can remove the ubiquitin added by ZNF598.

Despite our growing knowledge of the players and mechanisms involved in RQC, how the RQC process is regulated by cellular signaling pathways is less understood. RQC factors are present at sub-stoichiometric levels related to the ribosomal proteins. For example, yeast ZNF598 counterpart Hel2 was reported to be present at less than 1% of the abundance of ribosomes[42]. This low abundance of RQC factors relative to ribosomes creates a conundrum under stress conditions when the incidence of ribosome collision escalates. For example, amino acid deprivation, which leads to depletion of aminoacylated tRNAs, and alkylation and oxidation stresses, which damage RNA, are known to increase levels of ribosome collisions[17]. Although it has been shown that when RQC is overwhelmed under excess stress and ribosome collision, the integrated stress response[43,44] and ribotoxic stress response pathways[45,46] can be activated, leading to cell cycle arrest and apoptosis in cultured cells, little is known about how the RQC machinery responds to stress under physiological conditions and in intact animals.

Here, we examined the regulation of ZNF598 under stress in mammalian cells and in vivo in *Drosophila*. We found that ZNF598 undergoes regulatory K63-linked ubiquitination in a CNOT4-dependent manner and is upregulated upon mitochondrial stress. Overexpression (OE) of ZNF598 protects against mitochondrial stress

in cultured mammalian cells, *Drosophila* models of neurodegenerative disease, and patient cells by aborting stalled translation of mRNAs associated with mitochondrial outer membrane and removing faulty translation products causal of disease. Our results shed new light on the function of a previously unrecognized CNOT4-ZNF598 axis in mitochondrial and tissue homeostasis.

## Results

### Upregulation of ZNF598 protein level upon mitochondrial stress in cell culture

To investigate how cellular stress might impinge on the RQC process, we examined the expression of ZNF598 under various stress conditions. In cultured HeLa cells, we found that under conditions of serum starvation (EBSS), translational inhibition and autophagy induction with the mTOR inhibitor Torin1, or mitochondrial depolarization with CCCP, only CCCP caused a significant increase of ZNF598 protein level (Fig. 1a). Treatment of cells with the complex-I inhibitor rotenone also caused an increase of ZNF598, though to a lesser extent than observed in CCCP treated cells (Fig. 1b). The expression level of RACK1, another sensor of ribosome stalling and collision[24], was not changed by CCCP treatment (Fig. 1b). As the mRNA level of *ZNF598* was not changed under the mitochondrial damaging condition or other stress conditions (Fig. S1a), the regulation of ZNF598 likely occurred at the translational or post-translational level.

### ZNF598 is upregulated upon mitochondrial stress in vivo

We next tested whether the regulation of ZNF598 by mitochondrial stress holds true in vivo. For this purpose, we used *Drosophila* as our model system. Due to the lack of antibodies recognizing endogenous fly ZNF598 protein, we made use of a transgene expressing 3xHA-tagged ZNF598. We also examined 3xHA-tagged Rack1, Pelo, and ABCE1 proteins. Transgenes were expressed in the fly muscle using the *Mhc-gal4* driver and the *UAS-Gal4* system. Transgenic flies were fed for 7 days with fly food containing 250 μM rotenone or 100 μM CCCP. Western blot analysis of protein extracts prepared with dissected thoraxes showed that ZNF598-3HA was significantly increased upon CCCP treatment and moderately increased by rotenone treatment (Fig. 1c). In contrast, Pelo or Rack1 protein levels showed no obvious change (Fig. 1d, e). Consistent with previous findings[39], both rotenone and CCCP treatments led to a significant decrease in ABCE1 protein without affecting its mRNA levels (Fig. 1f, Fig. S1b). In transgenic flies with RNAi-mediated knockdown of PINK1, which causes mitochondrial dysfunction and stress[47], ZNF598 level was also increased (Fig. S1c), supporting that ZNF598 is being regulated by mitochondrial stress. *ZNF598* mRNA level was not altered by mitochondrial stress or PINK1 knockdown (Fig. S1d, e), again suggesting that the regulation of ZNF598 protein likely occurred at the translational or post-translational level.

We also tested if the ZNF598 protein level responds to mitochondrial stress in a mammalian setting. We used the well-established mitochondrial toxin 1-methyl-4-phenyl-1,2,3,6-tetrahydropyridine (MPTP) to induce mitochondrial stress in the mouse brain[48]. MPTP is a prodrug to the neurotoxin MPP+, which can be selectively taken up by dopaminergic neurons where it inhibits complex-I activity. We found that compared to RACK1, the level of ZNF598 in the mitochondrial fraction of striatum was moderately increased after MPTP treatment (Fig. 1g). The level of the cytosolic ribosomal protein RPS6 was also increased in the striatum mitochondrial fraction (Fig. 1g), suggesting accumulation of cytosolic ribosomes on the mitochondria surface upon mitochondrial stress, consistent with our previous findings[49,50]. The more robust accumulation of RPS6 than ZNF598 is consistent with ZNF598 and other RQC factors being sub-stoichiometric to ribosomes. The level of ANKZF1, which is homologous to the mitochondrial stress response protein Vms1 in yeast[51], was also increased in the mitochondrial fraction from striatum after MPTP treatment (Fig. 1g). It is worth

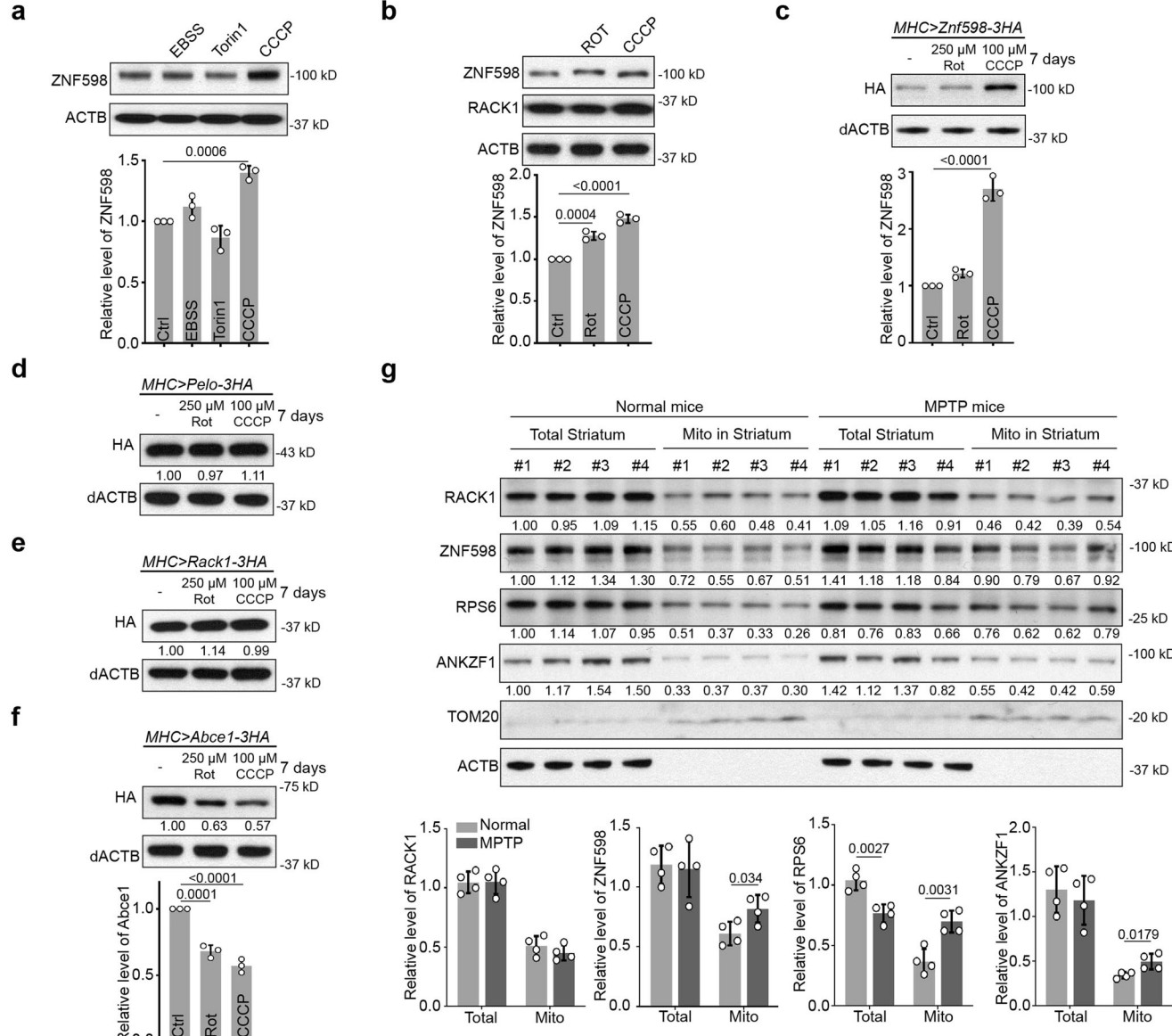

**Fig. 1 | ZNF598 is upregulated under mitochondrial stress conditions in vitro and in vivo. a** Western blot analysis and quantification of ZNF598 protein level in HeLa cells treated with EBSS (16 h), Torin1 (0.5 μM, 24 h), and CCCP (10 μM, 24 h). **b** Western blot analysis and quantification of ZNF598 protein level in HeLa cells treated with rotenone (5 μM, 24 h) or CCCP (10 μM, 24 h). Western blot analysis of HA-tagged ZNF598 (**c**), Pelo (**d**), Rack1 (**e**), and ABCE1 (**f**) proteins in the muscle tissue of flies treated with rotenone (250 μM) or CCCP (100 μM) for 7 days. Values under the blot show normalized protein levels relative to untreated sample in this figure and all subsequent ones. **g** Western blot analysis of the indicated proteins in total and mitochondrial fractions isolated from striatum tissues of normal and MPTP-indued mice. Data are representative of at least three biologically independent experiments (mean ± SD), $n = 3$ in (**a**–**c**), and (**f**), $n = 4$ in (**g**). Representative blot is from three independent experiments with similar results in (**a**–**f**). $p$ values in (**a**–**c**), and (**f**) were calculated by one-way ANOVA (Tukey's test). $p$ values in (**g**) were calculated by unpaired two-tailed $t$ test. Source data are provided as a Source Data file.

noting that since $MPP^+$ selectively accumulates in DA neurons and our western blot analysis was performed on whole striatum tissue, our observed changes in ZNF598 and ANKZF1 levels upon MPTP treatment may underestimate actual changes in brain cells actually experiencing MPTP-induced mitochondrial damage.

## ZNF598 protects cells against mitochondrial stress and other stress stimuli

To examine ZNF598 function in mammalian cells, we used CRISPR-Cas9 to knock out its expression. The knockout efficiency was confirmed by western blot analysis (Fig. S2a). Next, we challenged control, ZNF598 KO, and ZNF598 overexpressing cells with various stressors, including the mitochondrial stressors rotenone and CCCP, and the ER

stressors thapsigargin and Brefeldin A (BFA). ZNF598 KO cells were more sensitive to these stressors as measured with the CCK8 cell viability assay, and ZNF598 OE protected cells against the mitochondrial toxins (Fig. 2a, Fig. S2b). The fact that ZNF598 KO cells were sensitive to mitochondrial and ER stresses may reflect a multifaceted role of ZNF598 in regulating the function of diverse organelles or the intricate physical as well as physiological connections between these organelles such that defects in mitochondrial function may affect ER robustness.

We next tested the effect of ZNF598 on stress response in vivo. Transgenic flies overexpressing ZNF598 or with ZNF598 knockdown by RNAi were subjected to various stress conditions, including mitochondrial stress, ER stress and starvation. We found that ZNF598 OE significantly rescued CCCP- and starvation-induced ATP reduction

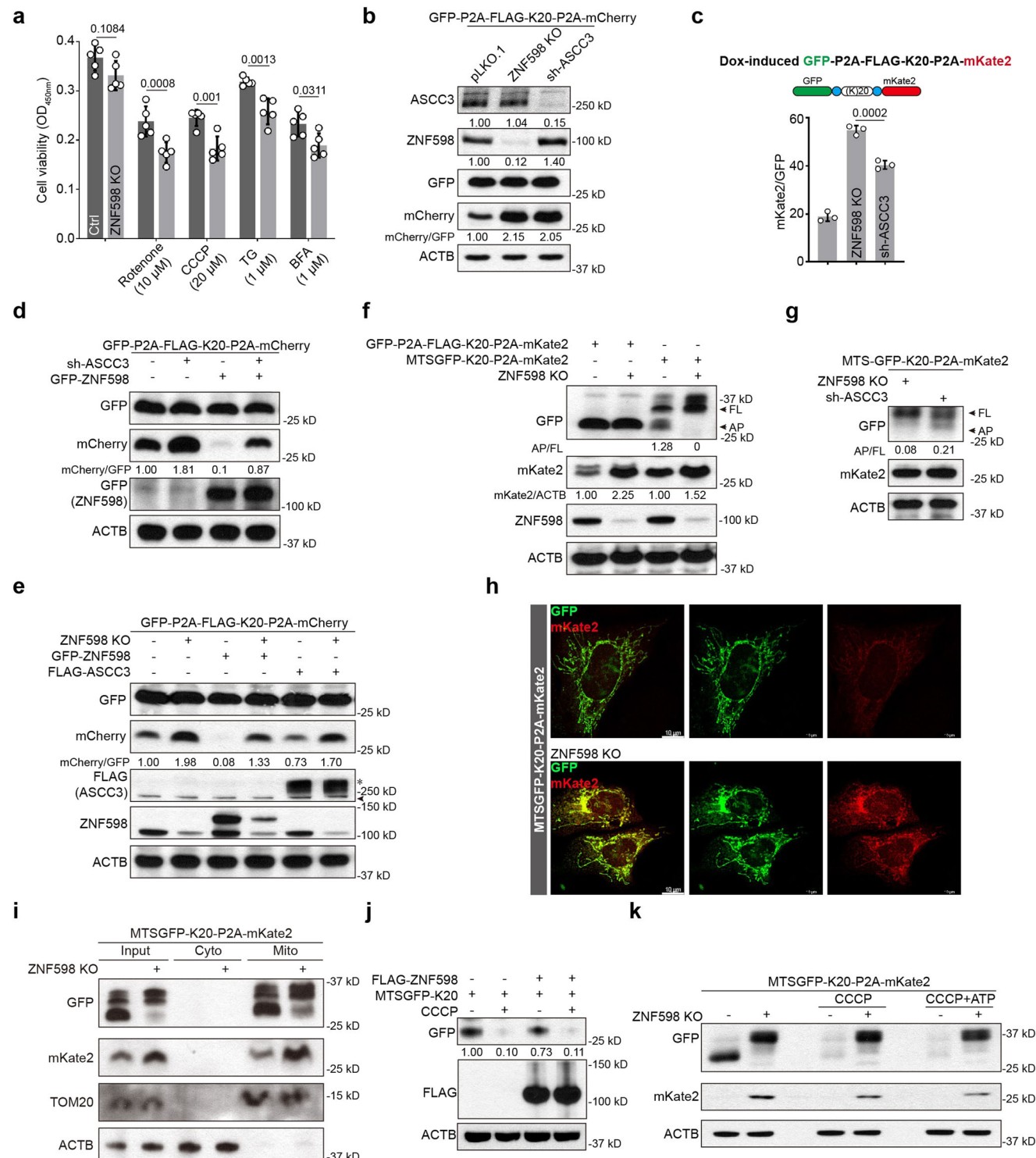

(Fig. S2c), supporting that ZNF598 helps maintain mitochondrial function during cellular stress.

## ZNF598 regulates quality control of stalled translation of mRNAs encoding mitochondria-targeted proteins

We next tested the mechanism of ZNF598 function in stress response. To assess the effect of silencing ZNF598 on translation, we used a GFP-P2A-Flag-K20-P2A-mCherry reporter[23], in which GFP, Flag-tagged 20 consecutive lysine codons (AAA, K20), and mCherry are used to monitor overall mRNA translation, translational stalling, and readthrough of the stalling site to synthesize the red fluorescent mCherry,

respectively. The self-cleaving P2A peptide sequence allows each reporter to be independent markers of translation. ZNF598 KO resulted in the removal of arrested translation at the Flag-K20 stall site, allowing increased synthesis of full-length mCherry (Fig. 2b). GFP expression was not significantly affected. Knocking down ASCC3, a helicase in the ASC-1 complex that selectively dissociates the leading ribosome of a collision, allowing trailing ribosome to continue translation, had similar effect (Fig. 2b). In cells expressing Dox-inducible stalling reporter, the fluorescence intensity of GFP and mCherry measured by flow-cytometry showed that ribosome readthrough as indicated by the mCherry/GFP ratio was higher in ZNF598 KO and

**Fig. 2 | ZNF598 regulates quality control of stalled translation of mRNAs encoding mitochondria-targeted proteins. a** Cell viability assay of control and ZNF598 KO HEK293T cells treated with rotenone, CCCP, TG and BFA for 24 h. **b** Immunoblots of GFP-P2A-Flag-K20-P2A-mCherry stall reporter expression in control (pLK0.1), ZNF598 KO, and ASCC3 KD HEK293T cells. **c** Flow cytometry analysis of GFP and RFP in normal, ZNF598 KO, and ASCC3 KD HEK293T cells expressing Dox-inducible GFP-P2A-Flag-K20-P2A-mKate2 after 1 µg/ml Dox incubation for 24 h. **d** Immunoblots showing effect of GFP-ZNF598 OE on GFP-P2A-Flag-K20-P2A-mCherry stall reporter expression in control (pLK0.1) and ASCC3 KD cells. **e** Immunoblots showing effect of GFP-ZNF598 and FLAG-ASCC3 OE on GFP-P2A-Flag-K20-P2A-mCherry stall reporter expression in control and ZNF598 KO cells. Asterisk indicates FLAG-ASCC3, and arrow indicates FLAG-Cas9 protein. **f** Immunoblots showing expression of GFP-P2A-Flag-K20-P2A-mCherry cytosol stall reporter and MTSGFP-K20-P2A-mKate2 mitochondrial stall reporter in control and ZNF598 KO cells. The ratio of arrested protein (AP) to full-length protein (FP) and mKate2 to actin were shown. **g** Immunoblots of MTS-GFP-K20-P2A-mKate2 stall reporter expression in ZNF598 KO, and ASCC3 KD HEK293T cells. For anti-GFP blot, the ratio of arrested GFP-K20 protein (AP) to full length protein (FP) is shown. **h** Immunostaining of GFP and mKate2 in normal and ZNF598 KO U2OS cells after transfecting with MTSGFP-K20-P2A-mKate2 for 24 h. **i** Immunoblots of GFP and mKate2 in mitochondria isolated from normal and ZNF598 KO HEK293T cells after transfecting with MTSGFP-K20-P2A-mKate2 for 24 h. **j** Effect of Flag-ZNF598 OE on MTS-GFP-K20 stall reporter expression in HeLa cells with or without CCCP treatment (6 hr). **k** Effect of ZNF598 KO on MTS-GFP-K20-P2A-mKate2 stall reporter expression in HeLa cell with or without CCCP or CCCP + ATP treatment. Data are representative of at least three biologically independent experiments (mean ± SD), n = 5 in (**a**), n = 3 in (**c**). Representative blot or image is from at least three independent experiments with similar results. p values in **a** were calculated by two-way ANOVA (Sidak's test). p values in (**c**) were calculated by one-way ANOVA (Tukey's test). Source data are provided as a Source Data file.

ASCC3 KD cells compared to control, with ZNF598 KO having a slightly stronger effect than ASCC3 KD (Fig. 2c, Fig. S2d). These results suggest that in the absence of sufficient ZNF598 or ASCC3 activity, the ribosomes may ignore the stall signal, or that the ribosomes may eventually bypass the stall site if they are not disassembled. OE of a GFP-ZNF598 construct only partially rescued mCherry expression in ASCC3 deficient cells (Fig. 2d). On the other hand, OE of Flag-ASCC3 did not significantly affect mCherry expression in ZNF598 KO cells (Fig. 2e), suggesting that ZNF598 may act downstream of ASCC3, but ZNF598-independent function of ASCC3 or ASCC3-independent function of ZNF598 may also exist.

Given the link between ZNF598 and mitochondrial stress response, we tested whether ZNF598 regulates the quality control of translation of mRNAs for nuclear-encoded, mitochondrial-targeted proteins, using a mitochondrial-targeted GFP reporter containing the K20 stall signal (MTS-GFP-K20-P2A-mKate2). As shown in Fig. S2e, the MTSGFP colocalized with mitochondria, and intriguingly, a significant portion of mKate2 also colocalized with mitochondria (Fig. 2h). Since mKate2 itself has not MTS, this result suggests that there is translation of the mitochondrial-targeted stall reporter on mitochondrial surface and co-translational import of the reporter proteins. This is consistent with co-translational import of mitochondrial proteins playing a critical role in cellular homeostasis[52], and a high frequency of co-translational import of nuclear-encoded mitochondrial proteins[53]. As with the cytosolically localized GFP-P2A-Flag-K20-P2A-mKate2 reporter, the MTS-GFP-K20-P2A-mKate reporter was also regulated by ZNF598 (Fig. 2f, g), and the effect of ZNF598 KO on the readthrough of the translational stall was comparable in the two reporter systems based on the relative mCherry signal (Fig. 2f). ZNF598 was seemingly more important than ASCC3 in regulating stalled translation of mRNAs encoding mitochondria-targeted proteins, as shown by more full length (FL) and less arrested protein of Flag-K20 in ZNF598 KO cells (Fig. 2g). The MTS-GFP-K20 and mKate2 proteins localized to mitochondria were increased in ZNF598 KO cells as shown by immunostaining (Fig. 2h) or immunoblots of isolated mitochondria (Fig. 2i), supporting a role of ZNF598 in regulating mitochondrial outer membrane-associated translation. On the other hand, OE of ZNF598 modestly decreased MTS-GFP-K20 expression (Fig. 2j).

### Mitochondrial stress activates ZNF598-mediated resolution of stalled translation

We further examined the effect of CCCP-induced mitochondrial stress on MTS-GFP-K20-P2A-mKate2 reporter expression. We found that CCCP treatment resulted in reduced expression of GFP-K20, and mKate2 was undetectable as in untreated cells (Fig. 2k). This CCCP effect was unlikely caused by mitochondrial energy deficit as it was not affected by the supplementation of ATP (Fig. 2k). The effect was likely due to increased ZNF598 expression as described earlier, and abortive termination of reporter translation by ZNF598. Indeed, ZNF598 KO effectively blocked the CCCP effect, resulting in the removal of arrested GFP-K20 and efficient translation of both full-length GFP-K20 and mKate2 (Fig. 2k).

The cyclic GMP-AMP synthase-stimulator of interferon genes (cGAS-STING) pathway senses cytosolic DNA and induces interferon signaling to activate the innate immune system. Translation stress and collided ribosomes were recently shown to serve as coactivators of cGAS[54]. Ribosome collision leads to cytosolic localization of cGAS, which preferentially interacts with collided ribosomes, and this ribosomal association stimulates cGAS activity. Using cytosolic cGAS localization as a proxy of its activation in response to ribosome stalling, we found that both anisomycin induction of ribosome collision and ZNF598 KO increased cytosolic cGAS and decreased nuclear cGAS signals, as previously reported[54]. However, while CCCP treatment antagonized the anisomycin effect on cGAS localization, it failed to do so in ZNF598 KO cells (Fig. S2f), supporting that ZNF598 mediates the effect of CCCP in resolving collided ribosomes and aborting stalled translation.

### Regulation by ZNF598 of stalled translation of mitochondrial outer membrane-associated *C-I30* mRNA in *PINK1* mutant flies

To test the in vivo function of ZNF598 in translational quality control of physiological substrates, we used the translation of mitochondrial complex-I subunit *C-I30* mRNA as a model. Previous studies showed that the translation of *C-I30* mRNA occurs on mitochondrial outer membrane[49], which is sensitive to mitochondrial stress or the loss-of-function of the mitochondrial quality control factor PINK1, mutations in which are associated with Parkinson's disease (PD)[55]. Mitochondrial dysfunction causes ribosome stalling at the canonical stop codon site of C-I30 due to the deficiency of translational termination and ribosome recycling activities and results in the formation of CAT-tailed C-I30-u species, which are toxic and contribute to PINK1 pathogenesis[50]. First, we tested whether ZNF598 and other RQC factors are localized to mitochondria in vitro and in vivo. U2OS cells were treated with CCCP or transfected with the MTSGFP-nonstop reporter, which would arrest ribosomes on the mitochondrial surface and stall translation of other mRNAs encoding mitochondria-targeted proteins[56]. As shown in Fig. 3a, colocalization between ZNF598 and mitochondria was increased after CCCP treatment, or when the MTSGFP-nonstop reporter was expressed. To detect the specific mitochondrial location of ZNF598, mitochondria isolated from normal and CCCP-treated cells were digested with Proteinase K in the presence or absence of Triton X-100. ZNF598 level was dramatically reduced by Proteinase K treatment in the absence of Triton X-100, suggesting that it was localized to mitochondrial outer membrane (Fig. 3b). In vivo, we found that ZNF598 and RQC factors Rack1, Pelo, Vms1/ANKZF1, and VCP showed various degrees of localization to mitochondria in the DA neurons of

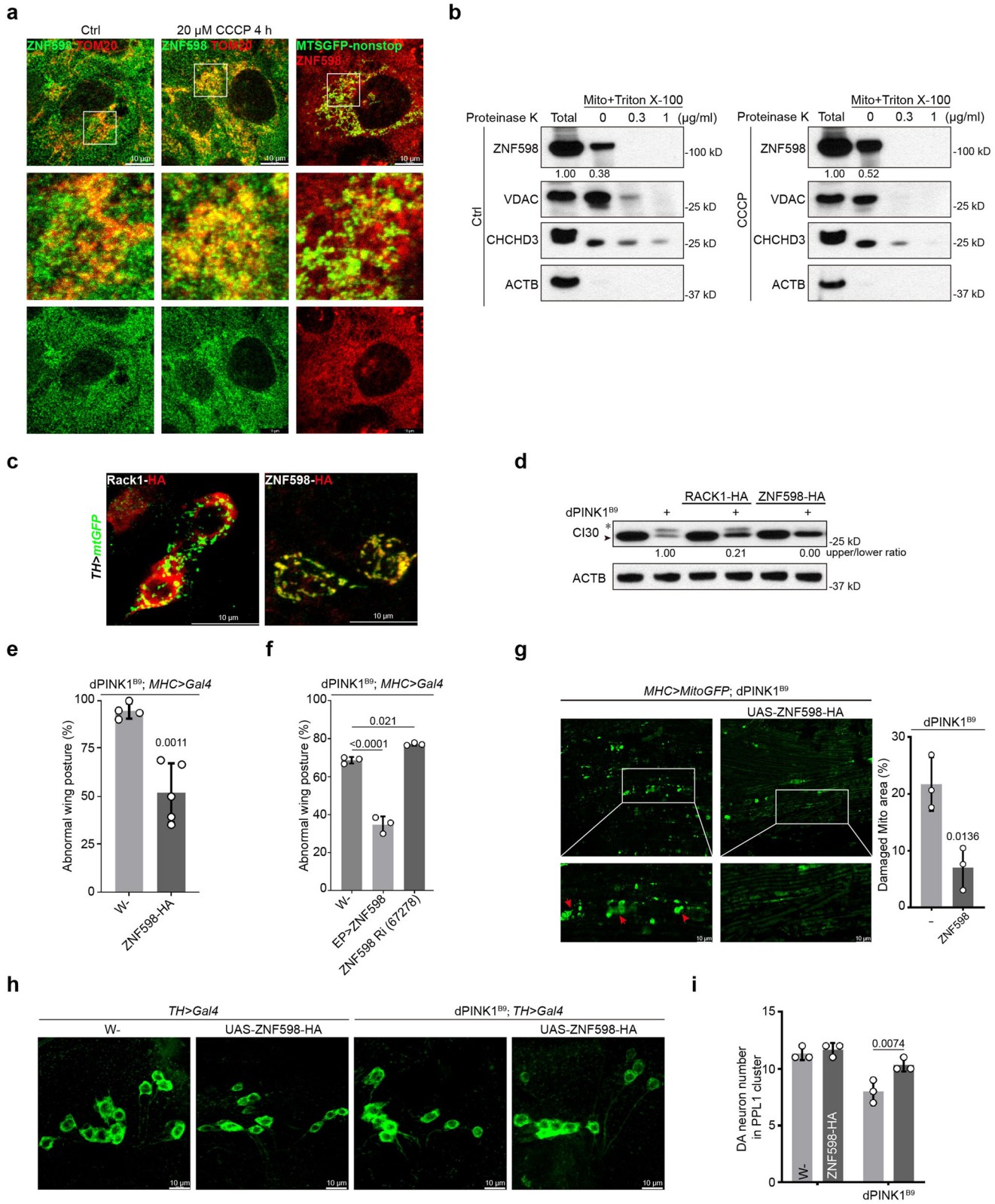

adult fly brains (Figs. 3c, S3a). Next, we tested the effect of OE of ZNF598 and the other RQC factors on C-I30-u formation. We found that ZNF598 had the strongest effect in eliminating C-I30-u (Fig. 3d). OE of the other RQC factors reduced C-I30-u level to varying degrees (Fig. S3b). For example, OE of HA-Rack1 had no obvious effect on C-I30-u level (Fig. 3d). These results demonstrate that ZNF598 responds

to mitochondrial stress in vitro and in vivo and plays a critical role in the quality control of mitochondria-associated ribosome stalling.

## ZNF598 OE rescues PD-related phenotypes *in PINK1* mutant
Consistent with the effect of ZNF598 on C-I30-u level, which strictly correlated with disease severity in *PINK1* mutant as shown previously[50],

**Fig. 3 | ZNF598 aborts stalled translation of mitochondrial outer membrane-associated *C-I30* mRNA and rescues mitochondrial and neuromuscular defects of *PINK1* mutant flies. a** Immunostaining of ZNF598 and TOM20 or GFP in U2OS cells under normal, CCCP treatment, and MTSGFP-nonstop OE conditions. **b** Immunoblots of ZNF598 in isolated mitochondria from normal and CCCP treatment cells, followed by proteinase K + Triton X-100 treatment. VDAC as out membrane, CHCHD3 as inner mitochondrial membrane, and ACTB as cytosolic protein markers. **c** Localization of HA-tagged early RQC factors ZNF598 and Rack1 to mitochondria of adult DA neurons. DA neuron mitochondria are marked with TH-Gal4-driven mito-GFP expression. **d** Western blot analysis showing effect of ZNF598 and Rack1 OE on CAT-tailed C-I30-u formation in *PINK1^B9* fly muscle. **e, f** Effect of ZNF598 OE (by UAS-ZNF598-HA or ZNF598-EP) or RNAi on abnormal

wing posture in *PINK1^B9* flies. **g** Effect of ZNF598 OE on mitochondrial morphology in *PINK1^B9* fly muscle. Red arrowheads mark aggregated mitochondria. Mito-GFP labels mitochondria. Bar graph shows quantification of damaged mitochondria. Effect of ZNF598 OE on DA neuron number in the PPL1 cluster of *PINK1^B9* adult brains (**h**). Bar graph shows data quantification (**i**). Data are representative of at least three biologically independent experiments (mean ± SD), n = 4 and 5 in (**e**), *n* = 3 in (**f, g, i**). Representative blot or image is from at least three independent experiments with similar results. *p* values in (**e**) and (**g**) were calculated by unpaired two-tailed *t*-test. p-values in (**f**) were calculated by one-way ANOVA (Tukey's test). *p* values in (**i**) were calculated by two-way ANOVA (Sidak's test). Source data are provided as a Source Data file.

OE of ZNF598 in the muscle effectively rescued the abnormal wing posture phenotype in *PINK1* mutant (Fig. 3e, f), which was caused by mitochondrial dysfunction-induced indirect flight muscle degeneration[47]. This was correlated with significant rescue of mitochondrial morphology as shown by the removal of defective, swollen mitochondria (Fig. 3g). We also examined the effect of ZNF598 OE on the PD-related DA neuron loss phenotype in *PINK1* mutant. ZNF598 effectively rescued the neuronal loss in the PPL1 cluster DA neurons in the *PINK1* mutant adult brain (Fig. 3h, i), concomitant with rescue of mitochondrial morphological defects (Fig. S3c). These data support the importance of ZNF598 in promoting mitochondrial and tissue homeostasis in a PD model.

## ZNF598 regulates the quality control of C9ALS-associated poly(GR) translation stalled on mitochondrial surface in *Drosophila*

We sought to further test the in vivo effect of ZNF598 on mitochondria-associated translational quality control. Expansion of G4C2 repeats in the *C9orf72* gene causes amyotrophic lateral sclerosis with frontotemporal dementia (C9ALS/FTD), one of the most common forms of ALS. The dipeptides translated from G4C2 repeat transcripts by unconventional translation, especially the arginine-containing poly(GR) and poly(PR), are considered disease-relevant toxic species[57–59]. We previously found that the translation of poly(GR) can occur on the mitochondrial surface, likely due to GR repeats mimicking mitochondrial-targeting signal, and ribosomes translating poly(GR) are frequently stalled, presumably caused by positively-charged Arg residues interacting with negatively-charged residues lining the ribosome exit channel[60]. Stalled translation of poly(GR) triggers CAT-tailing-like C-terminal extensions, which promote poly(GR) aggregation and toxicity[60]. OE of ZNF598 dramatically reduced the levels of both the CAT-tailed and non-CAT-tailed poly(GR) protein species, with the CAT-tailed species running as a smear slightly above the non-CAT-tailed species (Fig. 4a). This reduction of poly(GR) protein expression by ZNF598 OE is consistent with a recent report[61]. The reduction of poly(GR) protein expression by ZNF598 was also evident in immunostaining of muscle tissues (Fig. S4a). This was correlated with the rescue of the neuromuscular toxicity of poly(GR) as measured with wing posture defects in *Mhc>GR80* flies (Fig. 4b). RNAi of ZNF598 had opposite effects (Fig. 4c, d). The reduction of poly(GR) protein expression by ZNF598 correlated with its rescue of the mitochondrial morphology defect in *Mhc>GR80* fly muscle (Fig. 4e). These data support the importance of ZNF598 in promoting mitochondrial and tissue homeostasis in an ALS model.

## ZNF598 regulation of poly(GR) expression and toxicity in C9-ALS patient cells

We next tested ZNF598 function in regulating poly(GR) translation in a more physiologically relevant disease setting using C9ALS/FTD patient fibroblasts. OE of ZNF598 resulted in decreased poly(GR) expression, whereas ZNF598 RNAi had opposite effect in C9ALS/FTD patient cells (Fig. 4f). Consistent with this result, GR80 protein expression in HeLa

cells was significantly increased when ZNF598 or ASCC3 was silenced, but dramatically decreased when ZNF598 was overexpressed (Fig. 4g). Moreover, aggregates of GR80 were observed in the nuclei of ZNF598 or ASCC3 silenced cells (Fig. S4b). Furthermore, ZNF598 OE blocked the effect of ASCC3 silencing on GR80 expression (Fig. S4c), supporting that OE of ZNF598 may compensate for the effect of loss of ASCC3 on translation stalling.

As reported previously[62], C9ALS/FTD patient fibroblasts exhibited abnormal mitochondrial morphology and elevated mitochondrial membrane potential (MMP) and mitochondrial Ca²⁺ compared to cells from control subjects. The increased MMP is likely due to the alteration and tightening of cristae junctions caused by poly(GR), and resulting in impaired mitochondrial ion homeostasis[62]. OE of ZNF598 restored mitochondrial morphology (Fig. S4d), and reduced mito-Ca²⁺ level (Fig. 4h) and MMP (Fig. 4i) in C9-ALS/FTD patient fibroblasts. On the other hand, ZNF598 RNAi resulted in increased MMP (Fig. 4j). These results support the relevance of ZNF598 regulation of stalled poly(GR) translation to mitochondrial homeostasis and C9ALS/FTD pathogenesis.

## Poly(GR) interacts with early RQC factors ZNF598 and ASCC3, and with CNOT4

We further characterized the mechanisms involved in the co-translational quality control of poly(GR). RQC factor expression was measured in GR80 overexpressing cells, and the early RQC factor ASCC3, CNOT4 protein, and mRNA levels were found increased in FLAG-GR80 transfected cells (Figs. 5a, S5a). Intriguingly, in co-IP experiments, we found that FLAG-GR80 exhibited physical interaction with ZNF598, ASCC3, and CNOT4 (Fig. 5b). At least in the case of ZNF598, the interaction was preserved in RNase-treated extract, suggesting that the interaction was direct and not mediated by RNA (Fig. S5b). In the translational stalling reporter assays, GR80 was shown to inhibit the translational readthrough of the K20 stall, and the inhibitory effect was blocked in ZNF598 KO cells (Fig. 5c).

To examine the translation stalling of poly(GR), we performed polysome analysis. Compared to control cells, FLAG-GR80 transfected cells showed a moderate shift of the heavy polysome-associated Fragile X messenger ribonucleoprotein (FMRP) protein[63] to the lighter fractions (Fig. 5d). Although GR80 also caused a modest overall upregulation of FMRP level, we quantified the proportion of FMRP in each fraction relative to the total amount across the gradient. In fractions 4 and 5, which corresponded to collided ribosome fractions as indicated by the enrichment of ubiquitinated Rps3 and Rps10, lower MW species of Flag-poly(GR) corresponding to arrested translation products were detected. Moreover, CNOT4 was moderately enriched in these fractions (Fig. 5d). To distinguish collided ribosomes from normal ribosomes, RNase was applied to digest polysomes. Polysome profiling by A254 absorbance was applied to monitor ribosome collisions and increased RNase-resistant disomes in GR80 overexpressing cells was observed. As shown in Fig. 5e, fractions 4 and 5 from GR80 transfected cells corresponded to collided ribosome as also evidenced by the presence of ubiquitinated RPS3. ASCC3, ZNF598, and CNOT4

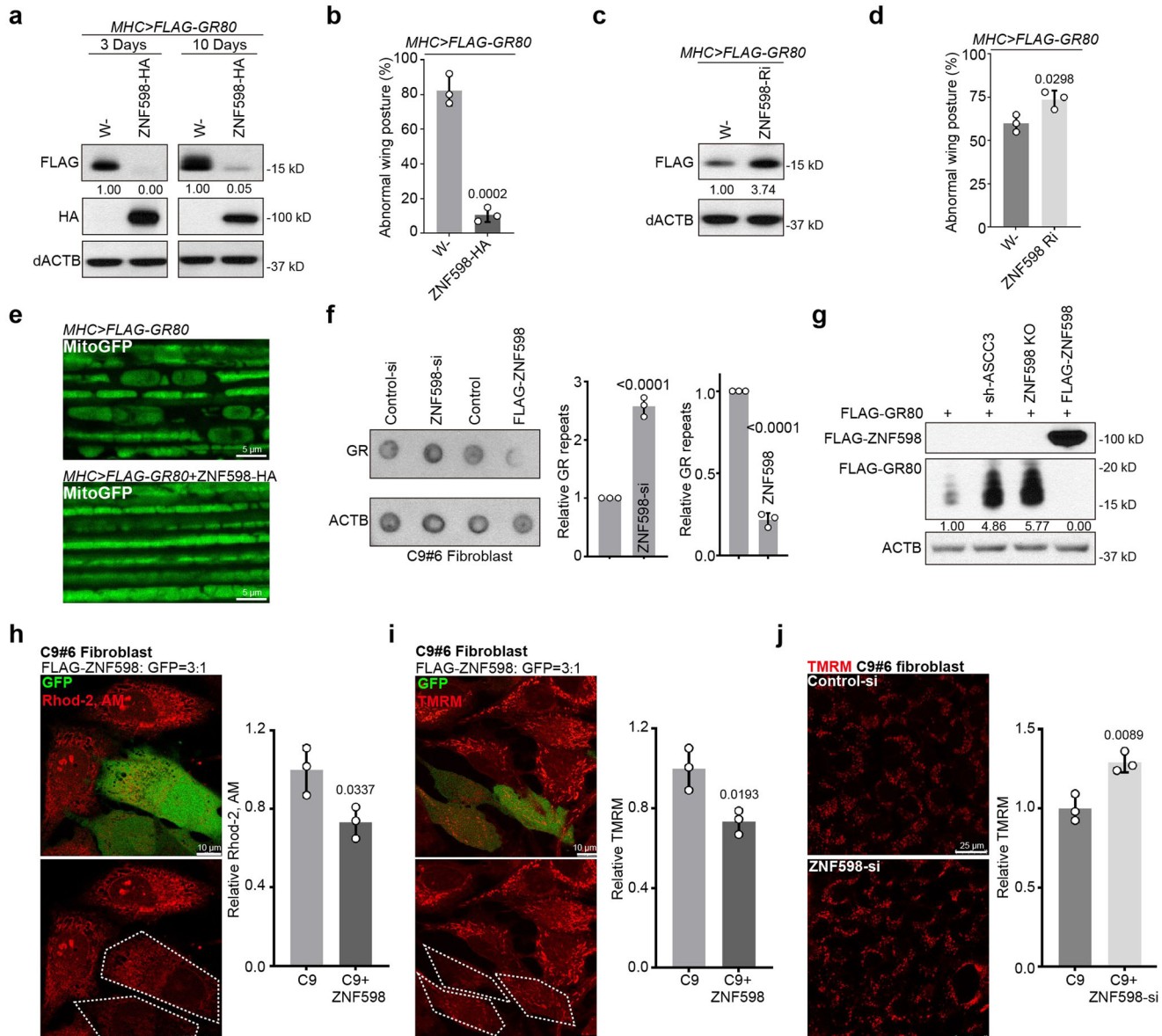

**Fig. 4 | ZNF598 regulates the quality control of stalled translation of C9ALS-associated poly(GR) and rescues poly(GR)-induced toxicity in *Drosophila* and patient cell models. a** Immunoblots showing the effect of ZNF598-HA OE on the level of Flag-tagged GR80 expressed in fly muscle. **b** Effect of ZNF598-HA OE on wing posture in *Mhc>GR80* flies. **c** Immunoblots showing effect of ZNF598 RNAi on the level of FLAG-tagged GR80 in *Mhc>GR80* flies. **d** Effect of ZNF598-RNAi on wing posture in *Mhc>GR80* flies. **e** Effect of ZNF598 OE on mitochondrial morphology in *Mhc > GR80* fly muscle, which normally exhibits vacuolated and disconnected mitochondria. **f** Dot blots showing effect of ZNF598 OE or RNAi on poly(GR) protein level in C9ALS patient fibroblasts. **g** Immunoblots showing effect of ASCC3 or ZNF598 silencing, or ZNF598 OE, on Flag-GR80 protein expression in HELA cells. Effect of ZNF598 on mitochondrial calcium stained by Rhod-2AM (**h**) and MMP stained by TMRM (**i**) in C9ALS fibroblasts. GFP was co-transfected with ZNF598 at 1:3 ratio so that virtually all ZNF598 transfected cells would be marked with GFP. Bar graphs show data quantification. **j** Effect of ZNF598 siRNA on MMP stained by TMRM in C9ALS fibroblasts. Bar graph shows data quantification. Data are representative of at least three biologically independent experiments (mean ± SD), *n* = 3. Representative blot or image is from at least three independent experiments with similar results. p-values were calculated by unpaired two-tailed *t*-test. Source data are provided as a Source Data file.

were enriched in these fractions compared to control cells. These data demonstrated the translational stalling of GR80 and suggested that CNOT4, in addition to ZNF598 and ASCC3, are recruited to handle the quality control of stalled ribosomes. Thus, not only the translation of poly(GR) is stalled that activates the RQC pathway, poly(GR) itself may also interact with RQC factors and interfere with the RQC process.

**K63-linked ubiquitination of ZNF598 upon mitochondrial stress**
We noticed that under mitochondrial damage conditions (rotenone and CCCP), ubiquitinated ZNF598 was increased (Fig. 6a). We tested if this might reflect autoubiquitination, a mechanism of E3 ligase self-regulation[64]. First, we found that GFP-ZNF598 and ZNF598-FLAG

exhibited robust interaction in co-IP assay (Fig. 6b), suggesting that ZNF598 is capable of self-association and thus autoubiquitination in *trans*. Next, we made use of E3-activity deficient mutants of ZNF598, ZNF598-CS[26] or ZNF598-C29A[24]. E3 activity was found to be essential for ZNF598 function in preventing the translational readthrough of K20 stall (Fig. 6c) and in aborting FLAG-GR80 translation (Fig. 6d). We also found that the level of ubiquitination was not much different between ZNF598 WT and ZNF598 CA mutant after CCCP treatment (Fig. 6e), suggesting that CCCP-induced ZNF598 ubiquitination may not be autoubiquitination and that other E3 ligase(s) may be involved.

To analyze CCCP-induced ubiquitination of ZNF598 further, we co-transfected Flag-tagged ZNF598 and HA-tagged ubiquitin and

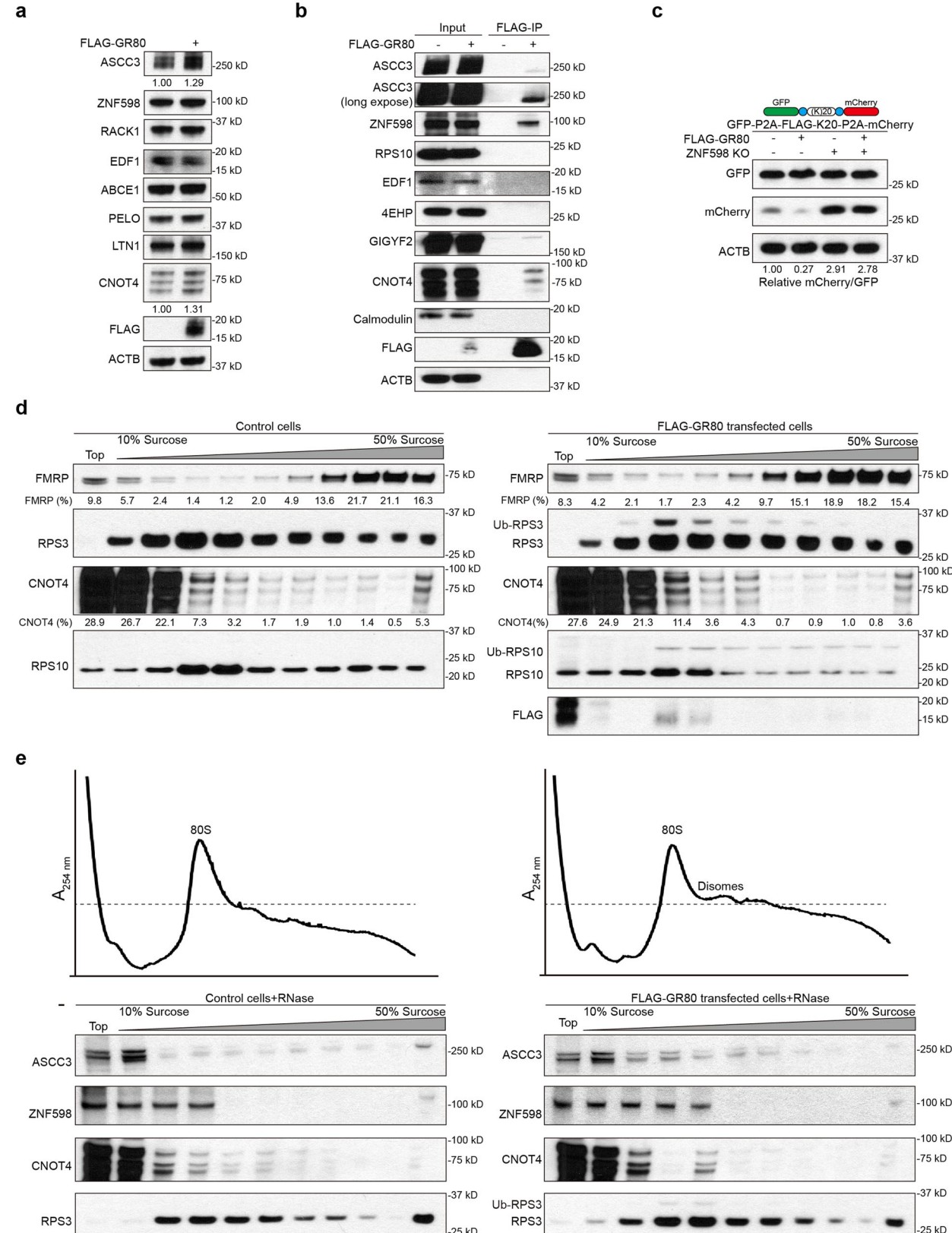

found that ZNF598 immunoprecipitated with anti-Flag under denaturing conditions was heavily ubiquitinated in CCCP-treated cells (Fig. 6f). Moreover, we co-transfected ZNF598 and mutant forms of ubiquitin and found that mutant Ub with K63 as the only available linkage (Ub-K63) recapitulated the effect of CCCP on ZNF598 ubiquitination, whereas mutant Ub with only K63 linkage blocked (Ub-K63R)

prevented the ability of CCCP to induce the formation of ubiquitinated ZNF598 species (Fig. 6g), suggesting that CCCP-induced mitochondrial stress promotes K63-linked ubiquitination of ZNF598. The association of this K63 ubiquitination with higher protein levels and higher activity of ZNF598 suggests that this event plays a regulatory and signaling function. Supporting this notion, transfection of HA-Ub-K63

**Fig. 5 | GR80 interacts with RQC factors and induces ribosomal stalling.**
**a** Western blot analysis of indicated proteins in HEK293T cells overexpressing
FLAG-GR80. **b** Immunoprecipitation analysis testing potential interaction between
FLAG-GR80 and RQC factors (ASCC3, ZNF598, RPS10, EDF1, 4EHP, GIGYF2,
CNOT4). Calmodulin serves as a negative control. **c** Immunoblots showing effect of
GR80 on translational readthrough of the GFP-P2A-FLAG-K20-P2A-mCherrry stall
reporter in normal and ZNF598 KO HEK293T cells. **d** Cell extracts from control (left
panels) or FLAG-GR80-overexpressing HEK293T cells (right panels) were fractio-
nated on 10-50% sucrose gradients. A total of 11 fractions were analyzed by SDS-
PAGE and immunoblotted with the indicated antibodies. **e** Cell extracts from
control (left panels) or FLAG-GR80-overexpressing HEK293T cells (right panels)
were incubated with RNase A (10 μg/ml) for 30 min and fractionated on 10-50%
sucrose gradients. Polysome profiles were derived from RNase-digested lysates of
HEK293T cells and GR80 overexpressing HEK293T cells. 11 fractions were analyzed
by SDS-PAGE and immunoblotted with the indicated antibodies. Representative
blot is from at least three independent experiments with similar results. Source
data are provided as a Source Data file.

partially inhibited the translational readthrough of K20 stall, and it
further enhanced the prevention of translational readthrough of
K20 stall by ZNF598 (Fig. 6h). In cell viability assays, the co-transfection
of HA-Ub-K63 significantly enhanced the resistance to CCCP-induced
cytotoxicity by ZNF598, whereas HA-Ub-K63R had no such
effect (Fig. 6i).

## The CNOT4-ZNF598 axis participates in quality control of stalled translation and mitochondrial stress response

We sought to explore the molecular mechanism involved in ZNF598
regulation. Given our data suggesting CNOT4 involvement in handling
stalled translation (Fig. 5), we focused on CNOT4. We found that like
ZNF598, CNOT4 was upregulated under mitochondrial stress condi-
tions (Fig. S6a), and in co-IP assays CNOT4 exhibited physical inter-
action with ZNF598 (Fig. S6b). CNOT4 OE did not significantly change
ZNF598 protein level, suggesting that CNOT4 might not regulate
ZNF598 stability (Fig. 7a). While the knockdown of CNOT4 resulted in
decreased ZNF598 polyubiquitination (Fig. 7b), increased poly-
ubiquitination of ZNF598 by CNOT4 was observed in the presence of
HA-Ub-K63 but not HA-Ub-K63R, supporting that CNOT4 promotes
K63-linked polyubiquitination of ZNF598 (Fig. 7c).

We next tested the effect of manipulating CNOT4 activity on
stalled translation. In vivo in the fly GR80 model, CNOT4 OE drama-
tically reduced Flag-GR80 translation, whereas CNOT4 RNAi had
opposite effect (Fig. 7d). In mammalian cells, CNOT4 attenuated the
translational readthrough of the K20 stall in the GFP-P2A-Flag-K20-
P2A-mCherry (Fig. 7e) and the MTS-GFP-K20 (Fig. S6c) reporter assays,
albeit with seemingly higher activity in the former assay. We further
tested the relationship between ZNF598 and CNOT4 in regulating
stalled translation using the GFP-P2A-Flag-K20-P2A-mCherry reporter
assay. The effect of CNOT4 OE in attenuating the translational read-
through of the K20 stall was completely blocked by the depletion of
ZNF598 (Fig. 7f, S6d), suggesting that ZNF598 may act downstream of
CNOT4 in handling stalled translation. Consistent with this notion, OE
of ZNF598 blocked the facilitation of translational readthrough of the
K20 stall by CNOT4 knockdown (Figs. 7g, S6e), and the co-expression
of CNOT4 and ZNF598 showed mild synergy in inhibiting the transla-
tional readthrough of the K20 stall (Fig. 7h).

Based on the information that the N terminus of ZNF598 is critical
for RQC[26], and that many of the lysine residues predicted by Phos-
phoSitePlus to be involved in ZNF598 ubiquitination are located at the
N-terminus, we made the ZNF598 KR mutant with the predicted
N-terminal ubiquitination sites (K80, 96, 97, 190, 204, 238, 263)
mutated to similarly positively changed Rs (Fig. S6f). The ubiquitina-
tion of ZNF598 KR mutant by CNOT4 was decreased compared to
ZNF598 WT (Fig. 7i), and it failed to block the readthrough of stalled
ribosomes either alone or together with CNOT4 (Fig. 7j). ZNF598 KR
mutant was less effective compared to ZNF598 WT in protecting
against mitochondrial stress (Fig. S6g). These data support the func-
tional significance of CNOT4-dependent ZNF598 ubiquitination in
RQC. In cell viability assays, CNOT4 OE protected against the cyto-
toxicity of CCCP-induced mitochondrial stress, an effect blocked by
the knockdown of ZNF598 (Fig. 7k). On the other hand, CNOT4 and
ZNF598 co-expression synergistically protected against CCCP-induced
toxicity (Fig. 7l). We further test the significance of ZNF598

ubiquitination to CNOT4 function. After knocking down endogenous
ZNF598 using shRNA against the 3′UTR, shRNA-resistant ZNF598 WT
and ZNF598 KR were introduced to rescue ZNF598. While ZNF598 WT
could decrease the readthrough of the K20 stall reporter in CNOT4
overexpressing cells with endogenous ZNF598 knocked down, ZNF598
KR had no effect (Fig.7m), suggesting that the CNOT4-ZNF598 axis
relies on the ubiquitination of ZNF598 to handle stalled ribosomes.
ZNF598 KR also failed to promote cell viability after CCCP treatment in
CNOT4 overexpressing cells with endogenous ZNF598 knocked down
(Fig. 7n), suggesting that CNOT4 depends on ZNF598 and its ubiqui-
tination to protect cells against mitochondrial stress. Together, these
data support a critical role of the CNOT4-ZNF598 axis in the quality
control of stalled translation and mitochondrial stress
response (Fig. S7).

## Discussion

The translational machinery is intimately linked to environmental
conditions, making ribosomes excellent candidates for sensors of the
cellular state and platforms for various signaling pathways that
respond to cellular changes. In particular, ribosome collision fre-
quency is considered a rheostat used by the cell to select the most
appropriate response to problems encountered during translation[17].
Under normal conditions or when stresses are manageable, cells may
use translation factors such as eIF5A to handle naturally occurring
stalls[65] or the RQC pathway to resolve infrequent collisions that result
from aberrant mRNAs. This typically result in resumption of transla-
tion. Under these conditions, maintaining RQC factors such as ZNF598
at sub-stoichiometry relative to ribosomes may be advantages to cells,
as too much ZNF598 activity may cause abortive translation of those
stalls that serve physiological purposes. The situation becomes more
complicated in more severe stress conditions when ribosome colli-
sions arise. Given the low abundance of RQC factors relative to ribo-
somes, it is conceivable that the RQC pathway will be overwhelmed
under these conditions, necessitating global stress responses or trig-
gering cell death[17]. Our results indicate that before cells succumb to
stress, there is an orchestrated RQC response that upregulates the
activity and abundance of ZNF598 and other RQC factors such as
CNOT4 and ANKZF1. Our results suggest that this upregulation occurs
mostly at the translational or post-translational levels at least in the
case of ZNF598. This upregulation on demand helps alleviate the
substoichiometry issue of ZNF598 and is important for maintaining
mitochondrial and tissue homeostasis under stress. Our results reso-
nate with the emerging concept that signaling on collided ribosomes
has consequences beyond that of just ribosome rescue and mRNA
quality control to encompass triggering of global stress responses[43,44],
including the cGAS-STING innate immune response[54], and cell fate
decisions[45,46]. We hypothesize that this ZNF598 regulation by upstream
stress signaling pathways may mechanistically link proteostasis,
mitochondrial homeostasis, and innate immune response, failures of
which constitute hallmarks of neurodegenerative diseases.

Our results show that ZNF598 protein level responds to mito-
chondrial stress, and that its upregulation promotes the quality con-
trol of stalled cytoplasmic ribosomes associated with mitochondrial
surface and the clearance of faulty translation products causal of dis-
ease in animal models of PD and ALS. The importance of ZNF598

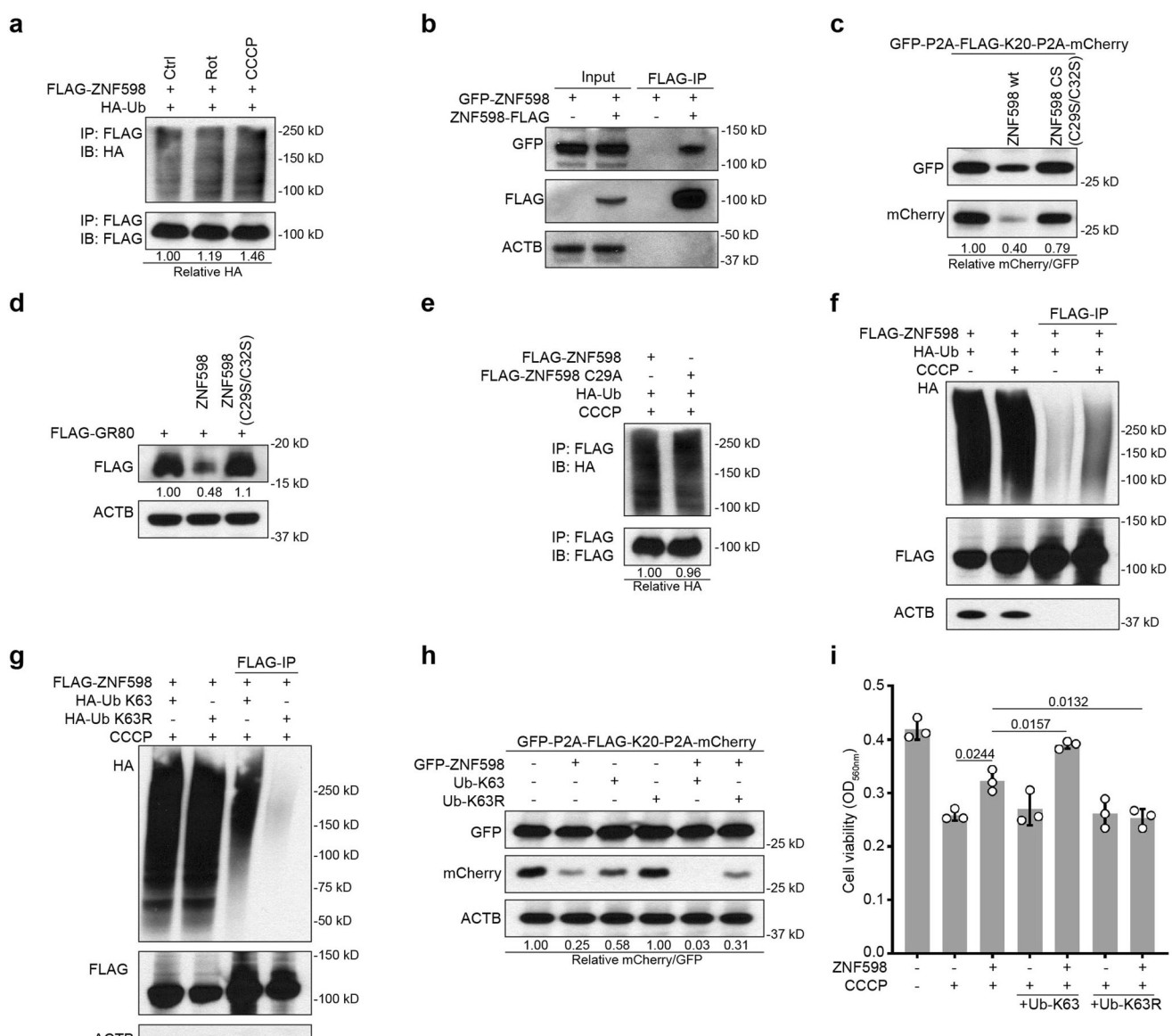

**Fig. 6 | ZNF598 undergoes regulatory K63-linked ubiquitination under mitochondrial stress. a** Immunoblots showing ubiquitinated ZNF598 in HEK293T cells co-transfected with FLAG-ZNF598 and HA-Ub. Cells were treated with 2 µM Rotenone or 10 µM CCCP for 24 h followed by denaturing FLAG-IP.
**b** Immunoprecipitation analysis of self-assembly of ZNF598 in cells co-transfected with GFP-ZNF598 and ZNF598-FLAG. **c** Effect of wild type ZNF598 and catalytically inactive mutant ZNF598 (ZNF598 CS) on GFP-P2A-Flag-K20-P2A-mKate2 stall reporter expression in HEK293T cells. **d** Effect of wild type ZNF598 and catalytically inactive ZNF598 (ZNF598 CS) on FLAG-GR80 expression in transfected HeLa cells.
**e** Immunoblots showing ubiquitination of ZNF598 WT and inactive ZNF598 mutant (ZNF598 CS) in HEK293T cells co-transfected with FLAG-ZNF598 and HA-Ub. Cells were treated with 10 µM CCCP for 24 h followed by denaturing FLAG-IP.
**f** Immunoblots showing effect of CCCP treatment (10 µM, 24 h) on ZNF598 ubiquitination in HEK293T cells co-transfected with Flag-ZNF598 and HA-Ub followed by denaturing IP and detection of Ub. The signals in the Flag-IP represent ubiquitinated ZNF598. **g** Immunoblots showing effect of CCCP treatment (10 µM, 24 h) on ZNF598 ubiquitination in HEK293T cells co-transfected with Flag-ZNF598 and HA-Ub-K63 or HA-Ub-K63R followed by denaturing IP and detection of Ub. All data are representative of at least three independent experiments. **h** Immunoblots showing effect of HA-Ub-K63 and HA-Ub-K63R OE on GFP-FLAG-K20-mCherry reporter expression in ZNF598 overexpressing HEK293T cells. **i** Cell viability measurement using CCK8 assay in HEK293T cells transfected with plasmids overexpressing ZNF598, Ub-K63, or Ub-K63R for 48 h followed by treatment with 20 µM CCCP (24 h). Data are representative of at least three biologically independent experiments (mean ± SD). Representative blot is from at least three independent experiments with similar results. $p$ values were calculated by one-way ANOVA (Tukey's test). Source data are provided as a Source Data file.

upregulation to mitochondrial and tissue homeostasis is consistent with previous studies implicating the crosstalk between cytosolic translation and mitochondrial function[52], and the particular involvement of the RQC pathway in maintaining mitochondrial homeostasis[50,66]. The nature of the mitochondrial signal that leads to ZNF598 ubiquitination remains to be determined. Mitochondrial outer membrane is known to be decorated with cytosolic ribosomes engaging in co-translational import of nuclear encoded mitochondrial proteins or proteins mistargeted to mitochondria, including the C-I30 and poly(GR) proteins studied here in the PD and ALS models, respectively[49,60]. Mitochondrial import is known to be sensitive to MMP and other parameters of mitochondrial function. Thus, defects in the co-translational import process may lead to translation stalling and ribosome collision under mitochondrial stress. It is also possible that mitochondrial dysfunction may lead to excessive ROS production, leading to damage of mitochondria-associated mRNA. Oxidizing

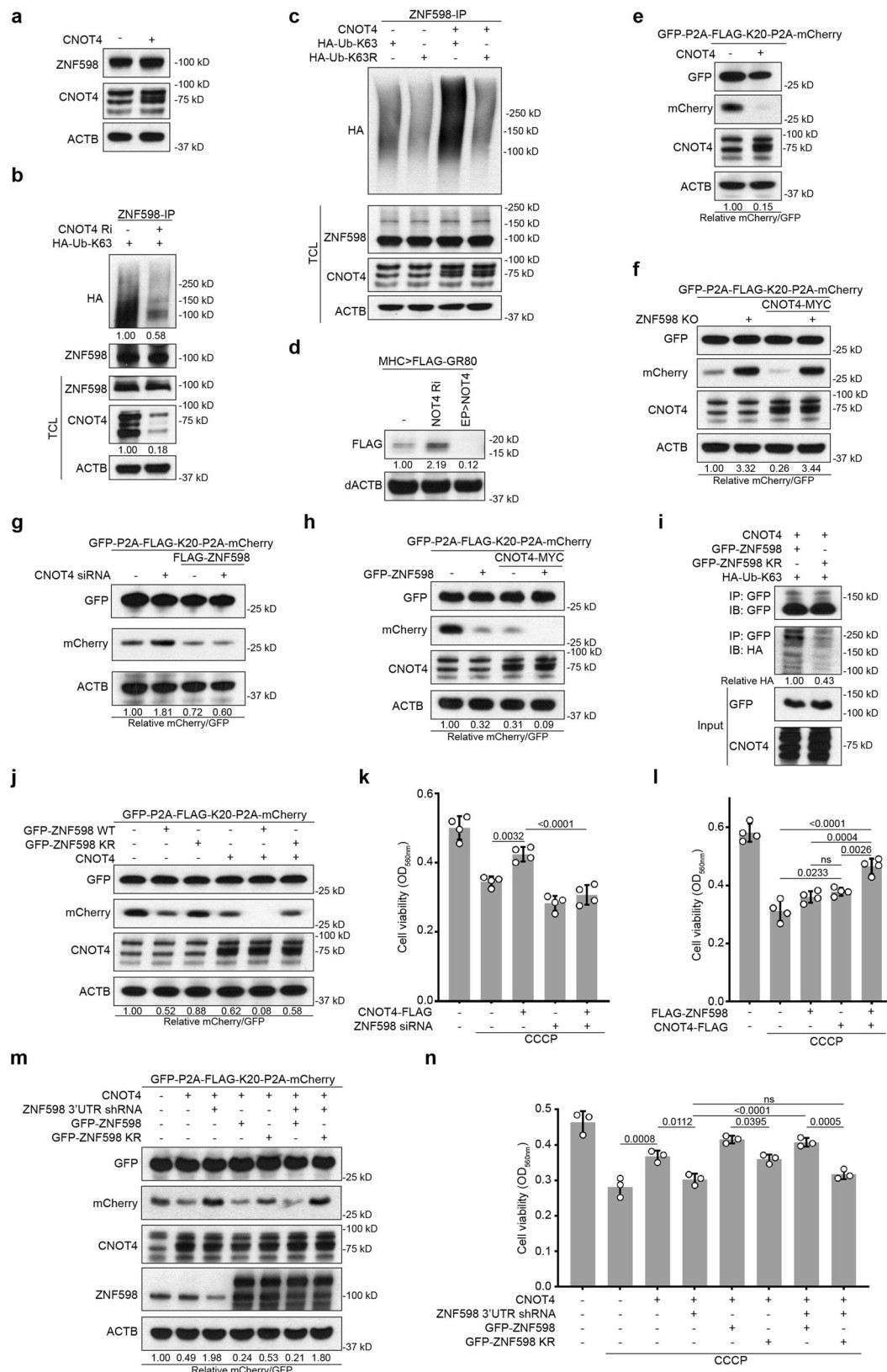

agents are known to modify the nucleobases of mRNAs, resulting in adducts such as 1-methyladenosine and 8-oxoguanosine, which inhibit tRNA selection by ribosomes and cause ribosome arrest[17]. In fission yeast, oxidative stress has been shown to cause a decrease in the levels of charged Trp-tRNA and thus ribosome stalling at Trp codons[67]. Whether similar events may occur in metazoans remains to be tested.

Other signals derived from mitochondrial stress, such as mitochondrial retrograde signals or mitochondrial unfolded protein response may also impinge on the RQC pathway. Future studies will test these possibilities.

Our results show that in response to mitochondrial stress, cellular RQC activity in aborting stalled translation was increased in a

**Fig. 7 | A CNOT4-ZNF598 axis regulates quality control of stalled translation and stress response. a** Immunoblots showing levels of ZNF598 in HEK293T cells with or without CNOT4 transfection. **b, c** Analysis of ubiquitination status of endogenous ZNF598 immunoprecipitated under denaturing condition from CNOT4-deficient HEK293T cells transfected with HA-Ub-K63 (**b**) or CNOT4 over-expressing HEK293T cells transfected with HA-Ub-K63 or HA-Ub-K63R (**c**). **d** Immunoblots showing effect of CNOT4 OE and CNOT4 knockdown on the level of FLAG-tagged GR80 expressed in fly muscle. **e** Immunoblots showing effect of CNOT4 OE on GFP-FLAG-K20-mCherry reporter expression. **f** Immunoblots showing effect of CNOT4 OE on GFP-FLAG-K20-mCherry reporter expression in ZNF598 KO HEK293T cells. **g** Immunoblots showing effect of CNOT4 knockdown on GFP-FLAG-K20-mCherry reporter expression in ZNF598 overexpressing HEK293T cells. **h** Immunoblots showing effect of CNOT4 OE on GFP-FLAG-K20-mCherry reporter expression in ZNF598 overexpressing HEK293T cells. **i** Immunoblots showing reduced ZNF598-KR ubiquitination by Ub-K63 when compared with ZNF598 WT.

**j** Immunoblots showing effect of CNOT4 OE on GFP-FLAG-K20-mCherry reporter expression in HEK293T cell overexpressing ZNF598 WT or ZNF598-KR. Cell viability measurement using CCK8 assay in HEK293T cells transfected with siRNA targeting ZNF598 and plasmid overexpressing CNOT4 for 48 h (**k**) or plasmid overexpressing ZNF598 or CNOT4 for 24 h (**l**). Cells were treated with 20 µM CCCP (24 h). **m** Immunoblots showing effect of ZNF598 WT and ZNF598 KR on GFP-FLAG-K20-mCherry reporter expression in CNOT4 overexpressing cells with endogenous ZNF598 knocked down. **n** Effect of ZNF598 WT and ZNF598 KR on cell viability in CNOT4 overexpressing cells treated with CCCP and with endogenous ZNF598 knocked down. Data are representative of at least three biologically independent experiments (mean ± SD), $n = 4$ in (**k, l**), $n = 3$ in (**n**). Representative blot is from at least three independent experiments with similar results. p-values were calculated by one-way ANOVA (Tukey's test). ns no significance. Source data are provided as a Source Data file.

ZNF598-dependent manner and that ZNF598 became poly-ubiquitinated in the process, apparently through ubiquitination by other E3 ligase(s). Further results reveal a previously unrecognized role of the E3 ligase CNOT4 in regulating ZNF598 polyubiquitination during the RQC process in response to mitochondrial stress. CNOT4 is known to respond to mitochondrial stress and it also promotes the ubiquitination of the ribosome recycling and RQC factor ABCE1[39]. CNOT4 was originally identified as a conserved component in the CCR4-NOT RNA quality control complex, but its importance in co-translational RQC[68], including regulatory ribosomal ubiquitination[69], is increasingly being recognized. Our result is consistent with findings in budding yeast showing that CNOT4 is involved in translational repression of problematic mRNAs causing ribosome staling, and that such activity helps maintain proteome integrity upon nutrient withdrawal[70]. Interestingly, a recent study in yeast described a role for CNOT4 in regulating mitochondrial outer membrane associated quality control of stalled translation of a mitochondrial matrix protein[71]. CNOT4, therefore, represents a potential link between mitochondrial stress-induced defective translation and RQC. Identification of upstream regulators of the CNOT4-ZNF598 axis will further our understanding of the regulation and function of the RQC pathway under physiological and stress conditions. Given the effect of PINK1 in recruiting RQC factors, including CNOT4 to mitochondrial outer membrane during mitochondrial stress[39], the effect of ZNF598 OE in rescuing *PINK1* mutant phenotypes, and the regulation of ZNF598 protein level by PINK1 as reported here, genes in the PINK1 pathways are candidate upstream regulators. In this respect, the E3 ligase Parkin is a good candidate, as Parkin is known to regulate mitochondria-associated C-I30 translation[49] and carry out K63-linked ubiquitination of substrates[72]. Previous studies also implicated a non-canonical Notch signaling pathway in regulating RQC[60]. It would be interesting to test a possible role of non-canonical Notch signaling in regulating the CNOT4-ZNF598 axis. Finding upstream regulators of the CNOT4-ZNF598 axis in response to mitochondrial stress will offer new insight into the regulation of RQC and help decipher how defects in this process may contribute to the pathogenesis of neurodegenerative diseases and other disorders.

## Methods

This research complies with all relevant ethical regulations. All procedures related to the handling, care, and the treatment of mice in this study were performed according to guidelines approved by the Laboratory Animal Care Committee of Stanford University following the guidance of the Association for Assessment and Accreditation of Laboratory Animal Care (AAALAC).

### *Drosophila* genetics

All *Drosophila* stocks were maintained at 25 °C on standard food incubators with a 12 h light:dark cycle. The fly stocks were obtained from the following sources: UAS-mito-GFP (William Saxton, University of California at Santa Cruz, Santa Cruz, CA), *PINK1^B9^* (Jongkeong Chung, Seoul National University, Seoul, Republic of Korea), UAS-FLAG-GR80 (Fenbiao Gao, UMass Medical School); The UAS-Rack1-3HA, -Pelo-3HA, -ZNF598-3HA, -ABCE1-3HA, -VCP-3HA, -Vms1/ANKZF1-3HA flies were purchased from FlyORF. The ZNF598 EP line, and ZNF598 RNAi fly lines were purchased from the Bloomington *Drosophila* Stock Center. Other stocks were generated in our laboratory. Fly culture and crosses were performed according to standard procedures. Adult flies were generally raised at 25 °C and with 12/12 h dark/light cycles. Fly food was prepared with a standard receipt (Water, 17 L; Agar, 93 g; Cornmeal, 1716 g; Brewer's yeast extract, 310 g; Sucrose, 517 g; Dextrose, 1033 g).

### Wing posture assays

Around 10–20 male flies were transferred to a clean plastic vial. To assay wing posture, cohorts of flies raised at 25° at the indicated ages were visually inspected for straight, held-up, or droopy wing postures. The number of flies with normal (straight) or abnormal (held-up or droopy) wing postures was counted and quantified as the percentage of the total number of flies.

### Western blot and dot blot assays

Cultured cells and fly thorax muscles were lysed in lysis buffer (50 mM Tris-HCl pH7.4, 150 mM NaCl, 10% glycerol, 1% Triton X-100, 5 mM EDTA) containing protease/phosphatase inhibitor (B14001/B15001, Bimake). After centrifugation at $14,000 \times g$ (4 °C) for 15 min and denaturation by boiling in loading buffer, samples containing 30 µg proteins were separated by 12.5% SDS-PAGE and transferred onto PVDF membranes. To detect GR repeats in C9-ALS fibroblast, dot blot was applied as described before[62]. 10 µg proteins were dropped on the PVDF membrane directly. After blocking with 5% nonfat milk at room temperature for 1 h, membranes were incubated with primary antibodies at 4 °C overnight. Membranes were rinsed three times with TBST (0.1% Tween20 in TBS) for 10 min each and incubated with HRP-conjugated secondary antibody for 1 h. After being washed three times with TBST for 10 min each, the membrane was completely immersed with ECL substrate (Perkin Elmer, Beaconsfield, UK) for 1 min. Special cares were taken to ensure proper sample loading amounts and exposure times so that protein amounts and signal intensities were in the linear range. Finally, the membranes were exposed to autoradiography film in a darkroom. The intensity of immunoblot bands was quantified using Fiji/NIH ImageJ software. Quantitative analysis of protein levels was determined and is expressed as a ratio to β-actin. Relative expression level was quantified as a ratio to control samples.

### MPTP-induced PD mice model study

C57BL/6 mice (male, 10 weeks old, 23–25 g) were purchased from the Jackson Laboratory. Animal welfare and experimental procedures were carried out strictly in accordance with the related ethical regulations of

Stanford University. MPTP-PD mice model was conducted as described previously[73]. Adult male C57Bl/6 mice were administered intraperitoneally with MPTP (dissolved in PBS) in a final concentration of 25 mg/kg daily for five consecutive days and maintained for another 7 days before sacrifice. Striatum was dissected to isolate mitochondria. In brief, striatum tissue was manually homogenized in homogenization buffer (30 mM Tris-HCl pH 7.4, 225 mM mannitol, 75 mM sucrose, 0.5 mM EGTA, protease inhibitor and 0.5% BSA) on ice. Nuclei and unbroken cell debris were pelleted by centrifugation at $700 \times g$ for 5 min. The supernatant was collected and centrifuged at $6300 \times g$ for 10 min at 4 °C, and the pellet was taken as crude mitochondria fraction.

### Immunofluorescence microscopy
For immunostaining of fly thorax and brain tissues, flies were anesthetized using $CO_2$-enriched air, and thorax or brain tissues were dissected and fixed in 4% paraformaldehyde containing 0.3% Triton X-100 at 4 °C overnight. The thoraces were subdivided into small chunks. The brain was dissected from the head cuticle, with the surrounding trachea carefully removed. Samples were permeabilized with 0.5% Triton X-100 in PBS for 45 min before blocking.

The cells were fixed with 4% paraformaldehyde at room temperature for 15 min, permeated with 0.5% Triton X-100 for 15 min, blocked with 5% BSA for 1 h, and incubated with primary antibodies overnight. After washing three times with 0.1% Triton X-100 for 10 min each, the cells were incubated with secondary antibodies for 1 h at room temperature, washed three times with 0.1% Triton X-100, and finally mixed with mounting medium containing DAPI. The images were obtained with a Leica SP8 laser confocal microscope (Leica, Germany). The immunofluorescence intensity and distribution were further analyzed using Fiji/Image J.

### Plasmid construction and transfection
The GFP-P2A-FLAG-K(AAA)20-P2A-mKate2 construct was modified based on GFP-P2A-FLAG-K(AAG)20-P2A-mCherry (105689, Addgene), and inserted into pCW57-MCS1-P2A-MCS2 (Blast) (80921, Addgene). For MTS (Mitochondrial targeting sequence)-GFP-K(AAA)20-P2A-mKate2 construction, the GFP-P2A-FLAG fragment was removed from GFP-P2A-FLAG-K(AAA)20-P2A-mKate2 plasmid, and the GFP fragment and MTS sequence derived from the first 29 AA of cytochrome c oxidase subunit 8A (COX8A) were inserted into the digested construct by using Gibson Assembly HiFi Master Mix (A46627, Invitrogen). The GFP and mCherry signals were detected by Western blot or flow cytometry at the Stanford FACS facility.

Cells were cultured to 70–80% confluence in six-well plates before transfection. Lipofectamine 2000 reagent (2.5 μl) was diluted with 100 μl Opti-MEM and gently blended with 100 μl Opti-MEM containing 2.5 μg plasmid. The mixture was placed at room temperature for 5 min and then added to the cell culture medium for 24 h. The ZNF598, ZNF598-CS[26], ZNF598-C29A[24], Flag-GR80, and Flag-(GR)n-GFP[60], Flag-cNOT4, HA-Ub, HA-Ub-K63, and HA-Ub-K63R[39] plasmids were described before. The FLAG-ASCC3 plasmid[26] was kindly provided by Dr. Toshifumi Inada. The GFP-ZNF598 (#141191) and FLAG-ZNF598 (#106590) plasmids were purchased from Addgene. The GFP-ZNF598 KR mutant plasmid in which the K80, 96, 97, 190, 204, 238, and 263 residues are mutated to Rs was synthesized by GenScript.

### Cell culture conditions
Regular HeLa cells and HEK293T cells (ATCC) were cultured under standard conditions (1x DMEM medium, 5% FBS, 5% $CO_2$, 37 °C). C9ALS/FTD patient fibroblasts and matched control fibroblasts were described before (Kramer et al. 2016) and kindly provided by Dr. Aaron Gitler. Hela, HEK 293T, and fibroblast cell transfections were performed using Lipofectamine 3000 (cat#: L3000015, Invitrogen), and si-RNA knockdown experiments were performed using Lipofectamine RNAiMAX reagent (cat#: 13778150, Invitrogen), according to manufacturer's instructions.

### Gene knockdown and knockout in cell culture
To reduce ASCC3 expression, shRNA targeting ASCC3 (5-CTACTT-CAAAGGCGGTATATA-3) was synthesized and cloned into pLKO.1 vector. pRSV-Rev, pMDL-RRE, pVSV-G were mixed with pLKO. 1 at the ratio of 1:1:1:2 and transfected into HEK293T cells for 72 h, and the lenti particles in supernatant were concentrated by using PEG-8000. To knockout ZNF598, CRISPR/Cas9 was used. Three sgRNAs targeting ZNF598 (sg1: 5-CCGATGACACGTCGCACCGT-3; sg2: 5-AGGCGG-TAGGCCCAAGAAGG-3; sg3: 5-CTACTGCGCCGTGTGCCGCG-3) were synthesized and cloned into the CRISPR V2 vector. Sh-ZNF598 targeting the 3´UTR (TRCN0000073158) was ordered from Sigma Aldrich. Viruses were produced in HEK293T cells using psPAX2, pMD2.G and CRISPR V2, concentrated using PEG-8000, and divided into small aliquot and stored in −80 °C fridge. Cells were plated in six-well plates in medium with 10% FBS and 2 μg/ml polybrene and infected with 20 μl of virus. The medium was refreshed, and puromycin (2 μg/ml) was added for selection after 48 h.

### Co-IP assay
Cells were homogenized in lysis buffer. After centrifuging at 14,000 $g$ and 4 °C for 15 min, the supernatant was immediately transferred to new tubes. Protein concentration was measured using BCA kits. 10 μl M2-anti-FLAG beads were added into lysis buffer and incubated overnight at 4 °C on a rocker. Then, the tubes were centrifuged at 500 g for 5 min, the pellet was washed three times with pre-cooled PBST, and the beads were boiled in loading buffer. The supernatants were collected and subjected to Western blot analysis. For denaturing Co-IP, cells were lysed with lysis buffer (50 mM Tris-HCl, pH 7.4, 150 mM NaCl, 1 mM EDTA, 10% glycerol, 1% triton x-100) containing phosphatase and protease inhibitor cocktail and supplemented with 2% SDS. Cell lysates were sonicated and boiled for 5 min, followed by diluting the lysates to a final concentration of 0.1% SDS for immunoprecipitation. After centrifuging the mixture at 12,000 $g$ for 10 min, the supernatant was collected for immunoprecipitation. The antibodies used are as follows. ZNF598 (HPA041760, Sigma-Aldrich); RACK1 (sc-17754, Santa Cruz); HA (81290-1-RR, Proteintech); RPS6 (2217, CST); ANKZF1 (20447-1-AP, Proteintech); TOM20 (sc-19964, Santa Cruz); ACTB (sc-47778, Santa Cruz); ASCC3 (17627-1-AP, Proteintech); GFP (66002-1-Ig, Proteintech); mCherry (26765-1-AP, Proteintech); RFP (R10367, Invitrogen); FLAG (F1804, Sigma-Aldrich); VDAC1 (MABN504, Millipore); CHCHD3 (25625-1-AP, Proteintech); CI30 (ab14711, Abcam); GFP (ab13970, Abcam); GR repeats (23978-1-AP, Proteintech); EDF1 (A2283, Abclonal); PELO (10582-1-AP, Proteintech); LTN1 (28452-1-AP, Proteintech); CNOT4 (10541-1-AP, Proteintech); RPS3 (ab128995, Abcam); RPS10 (ab151550, Abcam); 4EHP (13755-1-AP, Proteintech); GIGYF2 (24790-1-AP, Proteintech); Calmodulin (10541-1-AP, Proteintech); FMRP (13755-1-AP, Proteintech).

### Analysis of polysomes in cells
One 10 cm plate of control or Flag-GR80 transfected cells at 80% confluency was used per condition. Cells were collected and spun for 5 min at 1200 rpm. Lysis was performed in 200 μl of 1×RNC lysis buffer [50 mM HEPES, pH 7.6, 100 mM KOAc, 5 mM Mg(OAc)$_2$] supplemented with 0.5% Triton X-100, 1x protease inhibitor cocktail, 1 mM DTT and RNase Inhibitor (K1046, ApexBio) for 15 min on ice. Cells were ruptured by passing through a 26G needle using a 1 ml syringe (20 passes). Lysates were sedimented by centrifugation for 15 min at 12,000 $g$ at 4 °C, the supernatant containing 200 μg total RNA was loaded onto a 10-20-30-40-50% sucrose gradient in 1×RNC buffer. Centrifugation was for 30 min at 55,000 rpm in the TLS 55 rotor (Beckman) at 4 °C using the slowest acceleration and deceleration settings. Eleven fractions were collected manually from the top of the gradient.

## RNase resistance polysome profiling

HEK293T cells were lysed in a buffer consisting of 20 mM Tris-Cl (pH 8.0), 150 mM KCl, 15 mM MgCl₂, 1% Triton X-100, 1 mM DTT, phosphatase and EDTA-free protease inhibitor cocktails. Cell lysates were clarified by centrifugation at 3000 $g$ for 10 min and treated with RNase A (10 µg/mL) for 30 min at room temperature. RNase activity was quenched by adding 200 $U$ of SUPERaseIN. For polysome profiling, a 10−50% sucrose gradient was prepared in 20 mM Tris-Cl (pH 8.0), 150 mM KCl, 15 mM MgCl₂ using the Gradient Master (BioComp instruments). Cell lysates prepared above were applied atop the sucrose gradients and centrifuged using an ultracentrifuge at 35,000 $g$ for 3 h at 4 °C. Polysomes were fractionated and absorbance monitored at 254 nm using the TriaxTM flowcell (BioComp Instruments). In all experiments, cell lysates prepared from untreated cells served as controls.

## RT-qPCR

Quantitative real-time PCR was performed using PowerUp™SYBR™ Green Master Mix (A25742, Invitrogen) in a total volume of 20 µl using Applied Biosystems™ StepOne™ Real-Time PCR System (Applied Biosystems) as follows: 95 °C for 30 s, 40 cycles of 95 °C for 5 s, and 60 °C for 30 s. All results were normalized to the expression of beta-actin, and relative quantification was calculated by the $2^{-\Delta\Delta Ct}$ method. Primers used are as follows:

Human *ZNF598*: (Forward: 5′-ACCGCTGCTCTACCAAGATG-3′, Reverse: 5′-GTACAATGCGTACACCTTTCCA-3′); huma *ABCE1* (Forward: 5′-TGGAAAGTACGATGATCCTCCT-3′, Reverse: 5′-GGTCCAAAATAGATCCCACTGTC-3′); human *CNOT4* (Forward: 5′- CCTGCATGTAGAAAGCCATATCC-3′, Reverse: 5′- GTACACTAGCCAAATGTTTGCG-3′); human *ASCC3* (Forward: 5′- AAGTGGGGCTGCATTTCTCTT-3′, Reverse: 5′- TCGCCATGTTCTTTTTCTGTCAT-3′); *dABCE1* (Forward: 5′- GGGACAGAACGGTATTGGCA-3′, Reverse: 5′- TTGACCAGCGCCTTCAGATT-3′); *dZNF598* (Forward: 5′- CAAATCGATCAGCAGCCGTG-3′, Reverse: 5′-CCGATGCTTAGCGGAGATGT-3′).

## CCK8 cell viability assay

The viability of cells was determined using Cell Counting Kit-8 (CCK8) assays (HY-K0301, MCE) with the number of viable cells being evaluated after 2 h in medium containing CCK8. The conversion of the tetrazolium salt WST-8 to formazan was measured at 450 nm using a plate reader.

## Mitotracker Red, Rhod-2AM, and TMRM staining

For mitotracker staining, cells were washed with PBS and loaded with 500 nM MitoTracker Deep Red (M22425, Invitrogen) for 30 min. Cells were washed with PBS twice to remove the extra dyes. The fluorescent images were captured by microscope Leica SP8, and intensity was measured by Fiji/Image J. For Rhod-2AM staining, cells were washed with PBS and loaded with 5 µM Rhod-2 AM (R1244, Invitrogen) in HBSS without calcium/magnesium for 1 h. Cells were washed with HBSS twice to remove the extra dyes. The fluorescent images were captured using a Leica SP8 confocal microscope, and intensity was measured by Fiji/Image J. MMP was measured by using Image-iT™ TMRM Reagent (I34361, Invitrogen) according to the manual of the manufacturer. Fluorescent images were captured by Leica SP8 Confocal Microscope.

## Drug treatment

In Fig. 1a, b, Hela cells were treated with EBSS for 16 h, Torin1 (0.5 µM), CCCP (10 µM), rotenone (5 µM) for 24 h. In Fig. S1a, Hela cells were treated with Antimycin A (10 µM) and Oligomycin (10 µM), CCCP (20 µM) for 24 h. For cell viability in Fig. 2a, HEK293T cells were treated with mitochondrial toxin rotenone (10 µM) or CCCP (20 µM), ER stress inducer thapsigargin (TG, 1 µM) and Brefeldin A (BFA, 1 µM) for 24 h. In K20 stall reporter and ubiquitin-related experiments, cells were

treated with CCCP (20 µM) for 6 h. For drug treatment in flies, flies were fed with food containing rotenone (250 µM), CCCP (100 µM), or thapsigargin (TG, 5 µM) for 7 days, and drug-containing fly food was changed every day. Flies were starved for 16 h with water only before treatment.

## ATP measurement

ATP was measured using an ATP Bioluminescence Assay Kit CLS II (11699709001; ROCHE) by following the manufacturer's protocol and normalized with each thorax. The luciferase activity is measured on Lumat LB 9507 (Berthold Technologies).

## Mitochondria digestion with proteinase

For proteinase K digestion, crude mitochondrial fraction isolated from one 10-cm dish of HEK293T cells was suspended in resuspension buffer and equally divided into three parts. Each part was incubated with different proteinase K (0, 0.3, 1 mg/ml) with 1% Triton X-100 for 15 min on ice. Mitochondria were pelleted by centrifugation at $13,000 \times g$ for 10 min, and analyzed by SDS-PAGE and western blotting.

## Statistical analysis

Data are expressed as mean ± SD. Unpaired two-tailed $t$ test, one-way ANOVA (Tukey's test) and two-way ANOVA (Sidak's test) were used for statistical evaluation. All statistical analyses were conducted using GraphPad Prism Software Version 9.0 (GraphPad Software Inc., La Jolla, CA). Cases in which $P$ values of <0.05 were considered statistically significant.

## Material availability

Materials used in this study are available upon request to the corresponding authors and upon signing Material Transfer Agreements with Stanford University where applicable.

## Reporting summary

Further information on research design is available in the Nature Portfolio Reporting Summary linked to this article.

# Data availability

Data are available within the article and supplementary information. Source data are provided as Source Data file with this paper. Source data are provided with this paper.

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

## Acknowledgements

We are grateful to Drs. O. Brandman, E. Bennett, T. Inada, M. Hegde, FB. Gao, S. Juszkiewicz for plasmids and antibodies; Drs. S. Birman, T. Littleton, J Chung, FB. Gao, the Vienna *Drosophila* RNAi Center, FlyORF, and the Bloomington *Drosophila* Stock Center for fly stocks; Stanford PAN facility for primer synthesis; The Axelrod, Bogyo, Lipsick, and Svensson Labs in the Department of Pathology, Stanford University School of Medicine for sharing reagents and equipment. Special thanks go to J. Gaunce and W. Jiao for maintaining fly stocks and providing technical support and members of the Lu lab for discussions. This work was supported by the NIH (R01NS084412, R01NS083417, and R01AR074875 to B.L.; R01GM115968 to O.B.) and the National Natural Science Foundation of China (No. 32270825 to S.L.).

## Author contributions

J.G. designed the study, performed the experiments, analyzed data, and wrote the manuscript. Y.L., Z.W., S.B., S.R., D.K., and R.O. performed experiments and analyzed data. S.L. conceived and supervised the C9ALS part of the study, performed experiments, and provided funding. O.B. provided resources and funding. B.L. conceived and supervised the entire study, performed experiments, wrote the manuscript, and provided funding.

## Competing interests

B.L. is a co-founder and serves on the Scientific Advisory Board of Cerepeut, Inc. The remaining authors declare no competing interests.
