## [Peer Review File · Nature Communications]

Stalled translation by mitochondrial stress upregulates a CNOT4-ZNF598 ribosomal quality control pathway important for tissue homeostasisREVIEWER COMMENTS

Reviewer #1 (Remarks to the Author):

In this study Geng et al describe a quality control response to mitochondrial stress that involves the ZNF598 E3 ligase and its ubiquitination by the NOT4 E3 ligase. The authors observe that mitochondrial depolarization by CCCP causes an increase in ZNF598 protein in cultured HELA cells as well as in *Drosophila* and in flies with an RNAi-mediated knockdown of PINK1 that causes mitochondrial stress. Similarly in mouse brain, MPTP that is uptaken by dopaminergic neurons where it inhibits complex I activity, causes a slight increase of ZNF598 and an increase of RPS6 in the striatum mitochondrial fraction. Mammalian cells with a CRISPR-Cas9 knock out of ZNF598, were more sensitive to several stressors including CCCP, and transgenic flies with overexpressed ZNF598 rescued CCCP-induced ATP reduction. In cells knockout of ZNF598 suppressed translation arrested at a K20 stalling sequence of a reporter allowing expression of the full-length reporter, an effect also seen by knocking down ASCC3 that dissociates the leading ribosome of a collision. This was also true of a reporter with a MTS for which at least some translation occurs on the mitochondrial outer membrane, whilst overexpression of ZNF598 modestly inhibited its expression. CCCP treatment reduced expression of this MTS reporter and this was suppressed by ZNF598 overexpression. C130 mRNA has stalling on the stop codon and amino acid extension upon mitochondrial dysfunction by PINK1 loss of function that is suppressed by ZNF598 overexpression, and RQC factors are detected at mitochondria. Overexpression of ZNF598 in muscle suppresses abnormal wing phenotypes, and aberrant mitochondrial morphology in the PINK1 mutant. It also suppresses neuronal loss in the PINK1 mutant adult brain with rescue of mitochondrial defects.

Translation of poly(GR) (ALS model) can occur on the mitochondrial surface and show stalling, promoting CAT-tailing and aggregation. ZNF598 overexpression reduces levels of both CAT-tailed and non-CAT tailed species, and it rescues neuromuscular toxicity of poly(GR), in contrast to knockdown of ZNF598. In patient fibroblasts, ZNF598 overexpression also decreased poly(GR) expression and restored mitochondrial morphology. In HeLa cells, GR80 expression was increased when ZNF598 or ASCC3 were silenced but decreased when ZNF598 was overexpressed (that dominated over ASCC3 silencing).

The authors suggest that Flag-poly(GR) is present in the same polysome fractions as collided ribosomes and ubiquitinated ribosome proteins, where also an increased presence of NOT4 can be detected, and they suggest increased expression of NOT4 upon Flag-GR80 expression and a co-immunoprecipitation of NOT4 with Flag-GR80. Moreover, Flag-GR80 inhibits translation through the K20 stalling sequence.

CCCP induces ubiquitination of ZNF598, still detectable if catalytically inactive. This is inhibited by K63R-ubiquitin. Overexpression of ubiquitin is additive with overexpression of ZNF598 to inhibit readthrough of the K20 stall. Finally, the authors show that knockdown of NOT4 reduces ZNF598 ubiquitination and increases readthrough the K20 stall but not if ZNF598 is overexpressed.

From all of these results the authors suggest that NOT4 and ZNF598 regulate quality control of stalled translation and the mitochondrial stress response.

General

The topic of this work is very exciting. It provides new connections of the RQC, in particular a relationship to mitochondrial stress. It also indicates that yet another player in the RQC, ZNF598, is a target for NOT4 ubiquitination, extending the relevance of NOT4 in this response, and in the mitochondrial stress response. It is timely, because it shows how translation elongation that we are only starting to understand as an important regulatory step of gene expression, is a key step of cellular homeostasis control. Despite being very exciting findings, what is missing in the manuscript as it stands is some model for what is exactly going on. A figure with a proposed model would help to integrate the different observations.

Specific

I am making a number of specific comments in order of appearance in the manuscript, not necessarily in order of importance

1) In the introduction, the choice of references is not always correct. It was Panasenko et al., 2012, who showed that eS7 was a target of NOT4 ubiquitination. The reference for the role of NOT4 in quality control (Dimitrova et al., 2009), suggesting that NOT4 ubiquitinates the nascent chain, was discredited later by the Joazeiro lab, and instead the more recent manuscript by the Inada lab in 2019 (doi: 10.15252/embj.2018100276) showed that in case of defective RQT complex NOT4 ubiquitination can be relevant. This is a more relevant citation.

2) The increase of ZNF598 in figure 1G is rather minor compared to that of RPS6 that is noted but not discussed or follow up on, and really interesting. Some speculation about the meaning of this should be discussed.

3) Which cells are used in Figure 2? This is not mentioned in text or figure legend.

4) What happens to the mRNA under all conditions where protein goes up or down? The fate of the mRNA in general is lacking to an understanding of the mechanism.

5) The quantifications of Figure 2B for Flag and mKate2 at least, seem really aberrant when compared to the observed signals.

6) The authors propose to use MTS-GFP-K20-P2A-mKate because some of it should be translated on the mitochondrial membrane. How do the authors know that this “portion” of the translation is what is regulated by ZNF598 and not the cytosolic one? How does this reporter give any additional information without analysis of the mitochondrial associated translation?

7) For Figure 2E, the authors suggest that overexpression of ZNF598 could partially block ASCC3 effect. I would argue that ZNF598 has an effect independent of ASCC3.

8) The relevance of the addition of the section on cGAS-STING and anisomycin for the story of the manuscript is really not clear

9) Similarly, the relevance of the stalled nascent chain in the nucleolus and then polysome fractionation is not at all clear.

10) What is FMRP?

11) The data in Figure 5C is not clear. It seems that ubiquitinated Rps3 is already visible in control cells, is this expected? Is there is so much ribosome collision that one sees this in control cells? Then the Flag signal in the gradient is very unconvincing and does not overlap with the fraction of most increased NOT4 signal.

12) The authors use both NOT4 (text on figure 5) and CNOT4 (figure 5).

13) The co-ip of NOT4 with Flag-GR80 is not super convincing since the negative control has nearly the same signal.

14) The major signal that is increased in Figure 6A seems to be a discrete species rather than the smear.

15) For the experiment to show that Flag-ZNF598 is ubiquitinated by HA-Ubi would be to do an HA-IP and reveal by Flag, rather than a Flag IP and reveal higher MW species by Flag (Figure 6E) and for Figure 6F, it is the same, it should be HA IP revealed by Flag, because the smear could be co-purifying proteins that are ubiquitinated

16) Figure 7A is not convincing, and it is overall not convincing that NOT4 levels are limiting and an effect can be seen by its overexpression. In Figure 7E, 7F, and 7H, the effect of overexpressing NOT4 seems very variable. It could be that overexpression of NOT4 itself is variable or that NOT4 works with other proteins (the CCR4-NOT complex for instance) and overexpressed alone will not have a major effect. Do the authors know if it works in isolation or not? Instead, the depletion (Figure 7B) is convincing and it seems that it is the discreet ubiquitin band rather than the smear that decreases. I would certainly put more emphasis of this.

17) For Figure 7J, I do not see a synergy but a rather modest additive effect.

18) In the discussion, the authors refer to yeast, but should mention "fission yeast".

19) Does NOT4 impact the stability of ZNF598 since it impacts its ubiquitination, or is it like for ABCE1, not impacting the stability?

Reviewer #3 (Remarks to the Author):

Review for the manuscript NCOMMS-23-02506 by Dr. Bingwei Lu and co-authors entitled "A NOT4-ZNF598 axis responds to mitochondrial stress to quality control stalled translation and maintain tissue homeostasis".

Translational stalling events that result in ribosome collisions induce Ribosome-associated Quality Control (RQC) to degrade potentially toxic truncated nascent proteins. For RQC induction, the collided ribosomes are first marked by the ZNF598 E3 ubiquitin ligase to recruit the RQT complex for subunit dissociation. ZNF598 functionally marks collided mammalian ribosomes by K63-linked polyubiquitination of uS10 for the trimeric hRQT complex-mediated subunit dissociation. Although, the triggering steps of RQC are well-characterized, however, the physiological regulation of RQC in general and ZNF598 in particular in multicellular settings is underexplored.

In this study, the authors demonstrated the results suggesting that ZNF598 undergoes regulatory K63-linked ubiquitination under mitochondrial stress (Figure 6), and the NOT4-ZNF598 axis regulates quality control of stalled translation and stress response (Figure 7). Based on these, they proposed that ZNF598 undergoes regulatory K63-linked ubiquitination in a NOT4-dependent manner and is upregulated upon mitochondrial stresses in mammalian cells and *Drosophila*. The proposal is potentially interesting. However, the quality of the results is not sufficient enough to support their proposal. In special, the quantification of the signals is crucial in this manuscript. The signals of the binds detected by chemiluminescence/autoradiography must be quantified using the calibration curve. The main conclusion of this manuscript must be clarified by showing the phenotype of the ZNF598 mutant defective in NOT4-mediated ubiquitination.

Comments:

1. The main conclusion in this manuscript depends on the precise quantification of the signal intensities on the western blots. The signals of the binds detected by chemiluminescence/autoradiography must be quantified using the calibration curve. It is mandatory to demonstrate the quantification of the signal intensities using the calibration curve in all figures.
2. Based on the results shown in Figure 7, the authors propose that NOT4-dependent K63-linked ubiquitination of ZNF598 is required for resistance to mitochondrial stress. However, there is no direct evidence to demonstrate that Not4 directly ubiquitinates ZNF598, and the K63-linked polyubiquitination of ZNF598 somehow affects cell viability. The ubiquitination defective ZNF598 mutant must be constructed by the identification of lysine residue sites of NOT4-mediated ubiquitination. Moreover, the *in vitro* ubiquitination of ZNF598 by NOT4 may clearly show the linkage specificity.
3. In Figure 5, the authors show the co-IPed RQC factors are co-IPed with the FLAG-GR80. Genetic and biochemical research has been clarifying the substrate specificity of ZNF598-mediated ubiquitination and ASCC3/ASCC2/TRIP4-mediated splitting of the collided ribosomes. It is necessary to demonstrate that ZNF598 and ASCC3 are recruited to the ribosome stalled during the translation of FLAG-GR80 mRNA. The immunoprecipitated samples with FLAG-GR80 should be subjected to sucrose gradients to identify the disome or trisomes. The ribosome proteins should be detected with the CBB stain of the IPed samples. Moreover, FLAG-GR80 reduces mCherry by an unknown mechanism in Figure 5F. One of the required control experiments is the measurement of GFP and mCherry in ZNF598-KO.
4. Figure 6H, the difference in the mCherry/GFP ratios between with or without HA-Ub K63 expression is relatively small (0.47, lane 2 vs 0.24, lane 4). It is mandatory to perform the control experiments with Ub-K63R mutants.

5. In Figures 7F-G, the ribosome stalling at K20 in NOT4-KO is attenuated but increased by over expression of Not4. The precise quantification of GFP and mCherry products is crucial, therefore the measurement of fluorescence intensities of GFP/mCherry with flow-cytometry is highly recommended.

6. In Figure 2B, the level of mKate2 an indicator of ribosome stalling differs different in ZNF598-KO and ASCC3-KO cells. In principle, the ZNF598 and ASCC3 contribute to the splitting of the collided ribosomes, it's better to explain the reason why ribosome stalling is reduced differently in these KO cells.

7. In Figure 2A, the cell viability differs between Ctrl and ZNF598-KO without any treatment. Since the effects of 'the none' in cell viability by ZNF598-KO are significant in comparison with CCCP treatment, the p-value of the none must be demonstrated.

Reviewer #4 (Remarks to the Author):

In this manuscript the authors investigate the role of the protein ZNF598 in translation stalling and mitochondrial stress. They use a combination of mammalian cell, Drosophila and mouse models combined with genetic manipulation of ZNF598 and other factors involved in ribosome-associated quality control. They find that ZNF598 expression is increased by mitochondrial stress, acting downstream of ASCC3. ZNF598 is localised to mitochondria in Drosophila neurons and overexpression of ZNF598 rescues the phenotypes in Pink1 mutant flies. ZNF598 overexpression also rescues the phenotypes in Drosophila models and patient cells expressing the G4C2 C9orf72 repeats. They also show that ZNF598 levels are regulated through ubiquitination induced by mitochondrial stress. This is an interesting study and reveals new mechanisms associated with mitochondrial stress and the link to translation stalling through ZNF598. The manuscript is well written and the figures are clearly presented. I have the following comments:

1. The authors propose that ZNF598 and the regulation of ribosome stalling in response to mitochondrial stress are localised to the mitochondrial outer membrane. Although the authors show that ZNF598 is localised to mitochondria in Drosophila neurons more evidence is required to support the proposed mechanism. The authors should analyse ZNF598 localisation in mammalian cells. They should also analyse whether ZNF598 is localised to the mitochondrial outer membrane, intermembrane space or mitochondrial matrix using, for example, cellular fractionation studies combined with protease treatment of isolated mitochondria.

2. Expression levels are quantified beginning in Figure 1 and throughout the manuscript without statistical analysis, e.g. Figure 1A-F. If these quantifications are representative of several experiments, then this should be stated where appropriate in the figures. Also, in the first Results paragraph it is claimed that CCCP causes a 'significant' increase in ZNF598 expression. This gives the impression that

statistical analysis has been performed. The increase should either be shown to be statistically significant, or this word should be removed here and elsewhere, e.g. p.7.

3. The GFP-P2A-Flag-K20-P2A-mKate2 reporter is used extensively but this was designed as a reporter of cytosolic translation. The authors suggest that at least some translation involving the reporter would occur on the mitochondrial outer membrane. Can this be expanded upon? Is it known what proportion of translation occurs on the mitochondrial outer membrane? It seems surprising that the reporter is so sensitive to changes in ZNF598 levels when this probably only regulates a small proportion of overall translation.

4. On p.12 it is stated that the effect of overexpression of ZNF598 and other RQC factors on C-130-u formation is tested. However, only data for HA-Rack1 is shown in Figure 3B. Data for the other RQC factors should be included, or this statement modified.

5. The experiment shown in Figure 4F should be analysed statistically with repeats.

6. It would be very helpful to include a model for the proposed mechanism of regulation of ribosome stalling by ZNF598 in response to mitochondrial stress in the figures.

Minor comments:

1. On p.8 it states that ZNF598 KO cells are more sensitive to all the mitochondrial stressors tested but the decrease in viability with BFA treatment is not significant, so this should be mentioned.

2. A citation should be included for the GFP-P2A-Flag-K20-P2A-mKate2 reporter on p.9. Also, a more details description of this reporter would be helpful.

3. Scale bars should be added to all immunofluorescence images.

Point-by-Point Response to Reviewer Comments on Nat Commun Manuscript NCOMMS-23-02506

Reviewer #1 (Remarks to the Author):

In this study Geng et al describe a quality control response to mitochondrial stress that involves the ZNF598 E3 ligase and its ubiquitination by the NOT4 E3 ligase. The authors observe that mitochondrial depolarization by CCCP causes an increase in ZNF598 protein in cultured HELA cells as well as in *Drosophila* and in flies with an RNAi-mediated knockdown of PINK1 that causes mitochondrial stress. Similarly in mouse brain, MPTP that is uptaken by dopaminergic neurons where it inhibits complex I activity, causes a slight increase of ZNF598 and an increase of RPS6 in the striatum mitochondrial fraction. Mammalian cells with a CRISPR-Cas9 knock out of ZNF598, were more sensitive to several stressors including CCCP, and transgenic flies with overexpressed ZNF598 rescued CCCP-induced ATP reduction. In cells knockout of ZNF598 suppressed translation arrested at a K20 stalling sequence of a reporter allowing expression of the full-length reporter, an effect also seen by knocking down ASCC3 that dissociates the leading ribosome of a collision. This was also true of a reporter with a MTS for which at least some translation occurs on the mitochondrial outer membrane, whilst overexpression of ZNF598 modestly inhibited its expression. CCCP treatment reduced expression of this MTS reporter and this was suppressed by ZNF598 overexpression. C130 mRNA has stalling on the stop codon and amino acid extension upon mitochondrial dysfunction by PINK1 loss of function that is suppressed by ZNF598 overexpression, and RQC factors are detected at mitochondria. Overexpression of ZNF598 in muscle suppresses abnormal wing phenotypes, and aberrant mitochondrial morphology in the PINK1 mutant. It also suppresses neuronal loss in the PINK1 mutant adult brain with rescue of mitochondrial defects.

Translation of poly(GR) (ALS model) can occur on the mitochondrial surface and show stalling, promoting CAT-tailing and aggregation. ZNF598 overexpression reduces levels of both CAT-tailed and non-CAT tailed species, and it rescues neuromuscular toxicity of poly(GR), in contrast to knockdown of ZNF598. In patient fibroblasts, ZNF598 overexpression also decreased poly(GR) expression and restored mitochondrial morphology. In HeLa cells, GR80 expression was increased when ZNF598 or ASCC3 were silenced but decreased when ZNF598 was overexpressed (that dominated over ASCC3 silencing).

The authors suggest that Flag-poly(GR) is present in the same polysome fractions as collided ribosomes and ubiquitinated ribosome proteins, where also an increased presence of NOT4 can be detected, and they suggest increased expression of NOT4 upon Flag-GR80 expression and a co-immunoprecipitation of NOT4 with Flag-GR80. Moreover, Flag-GR80 inhibits translation through the K20 stalling sequence.

CCCP induces ubiquitination of ZNF598, still detectable if catalytically inactive. This is inhibited by K63R-ubiquitin. Overexpression of ubiquitin is additive with overexpression of ZNF598 to inhibit readthrough of the K20 stall. Finally, the authors show that knockdown of NOT4 reduces ZNF598 ubiquitination and increases readthrough the K20 stall but not if ZNF598 is overexpressed.

From all of these results the authors suggest that NOT4 and ZNF598 regulate quality control of stalled translation and the mitochondrial stress response.

General

The topic of this work is very exciting. It provides new connections of the RQC, in particular a relationship to mitochondrial stress. It also indicates that yet another player in the RQC, ZNF598, is a target for NOT4

ubiquitination, extending the relevance of NOT4 in this response, and in the mitochondrial stress response. It is timely, because it shows how translation elongation that we are only starting to understand as an important regulatory step of gene expression, is a key step of cellular homeostasis control. Despite being very exciting findings, what is missing in the manuscript as it stands is some model for what is exactly going on. A figure with a proposed model would help to integrate the different observations.

We thank the reviewer for the thoughtful and overall positive comments. In Fig. S7 we have provided a model of the newly discovered CNOT4-ZNF598 axis in resolving stalled ribosomes on mitochondrial surface induced by mitochondrial stress.

Specific

I am making a number of specific comments in order of appearance in the manuscript, not necessarily in order of importance

1) In the introduction, the choice of references is not always correct. It was Panasenko et al., 2012, who showed that eS7 was a target of NOT4 ubiquitination. The reference for the role of NOT4 in quality control (Dimitrova et al., 2009), suggesting that NOT4 ubiquitinates the nascent chain, was discredited later by the Joazeiro lab, and instead the more recent manuscript by the Inada lab in 2019 (doi: 10.15252/embj.2018100276) showed that in case of defective RQT complex NOT4 ubiquitination can be relevant. This is a more relevant citation.

We thank the reviewers for pointing us to the right references. We have now chosen the indicated references.

2) The increase of ZNF598 in figure 1G is rather minor compared to that of RPS6 that is noted but not discussed or follow up on, and really interesting. Some speculation about the meaning of this should be discussed.

We thank the reviewer for the suggestion. We have now speculated on page 7 that “The more robust accumulation of RPS6 than ZNF598 is consistent with ZNF598 and other RQC factors being sub-stoichiometric to ribosomes.”

3) Which cells are used in Figure 2? This is not mentioned in text or figure legend.

We used HEK293T cells and U2OS cells in Figure 2. Cell type is now mentioned in all figure legend.

4) What happens to the mRNA under all conditions where protein goes up or down? The fate of the mRNA in general is lacking to an understanding of the mechanism.

The reviewer’s point is well taken. We have now measured the mRNA levels under the various conditions where proteins levels are changed. These data are presented in **Fig. S1a, 1b, 1d, 1e and Fig. S5a**. Our results show that the mRNA level of ZNF598 has no significant change in the various groups where the protein level is changed. The mRNA levels of ABCE1 and ZNF598 are not changed in GR80 overexpressing cells,

but CNOT4 and ASCC3 mRNAs are significantly changed in GR80 overexpressing cells. The underlying mechanisms will require future investigation.

5) The quantifications of Figure 2B for Flag and mKate2 at least, seem really aberrant when compared to the observed signals.

We thank the reviewer for pointing that out. In previous data, the FLAG quantification in ZNF598 KO and sh-ASCC3 is reversed. We repeat the experiment using GFP-P2A-FLAG-K20-P2A-mCherry and detect the GFP and mCherry signal using WB and flow cytometry (Fig. 2b, c and Fig. S2d). Given that the FLAG-K20 arrested product is subject to post-translation modifications and degradation and very unstable, we have removed the FLAG signal and use mCherry/GFP ratio as a key measure of ribosome stalling and readthrough.

6) The authors propose to use MTS-GFP-K20-P2A-mKate because some of it should be translated on the mitochondrial membrane. How do the authors know that this “portion” of the translation is what is regulated by ZNF598 and not the cytosolic one? How does this reporter give any additional information without analysis of the mitochondrial associated translation?

The reviewer’s point is well taken. As shown in Fig. S2e, the MTS-GFP colocalizes with mitochondria, and intriguingly, a significant portion of mKate2 also colocalizes with mitochondria (Fig. 2h). Since mKate itself has not MTS, this result suggests translation of the mitochondrial-targeted stall reporter on mitochondrial surface and co-translational import of the reporter proteins. The MTS-GFP-K20 and mKate2 proteins localized to mitochondria were increased in ZNF598 KO cells as shown by immunostaining (Fig. 2h) or immunoblots of isolated mitochondria (Fig. 2i), supporting the role of ZNF598 in regulating mitochondrial outer membrane associated stalled translation. We would also like to point out that ZNF598 regulates both cytosolic and mitochondrial-associated stalled translation, as the effect of ZNF598 KO on the readthrough of the translational stall was comparable in the cytosolically localized GFP-P2A-Flag-K20-P2A-mKate2 reporter and the MTS-GFP-K20-P2A-mKate reporter systems based on the relative mCherry signal (Fig. 2f).

7) For Figure 2E, the authors suggest that overexpression of ZNF598 could partially block ASCC3 effect. I would argue that ZNF598 has an effect independent of ASCC3.

We understand the reviewer’s point. On page 9 we rephased the text as follows: OE of a GFP-ZNF598 construct only partially rescued mCherry expression in ASCC3 deficient cells (Fig. 2d). On the other hand, OE of Flag-ASCC3 did not significantly affect mCherry expression in ZNF598 KO cells (Fig. 2e), suggesting that ZNF598 may act downstream of ASCC3, but ZNF598-independent function of ASCC3 or ASCC3-independent function of ZNF598 may also exist.

8) The relevance of the addition of the section on cGAS-STING and anisomycin for the story of the manuscript is really not clear

Conditions leading to collided ribosomes such as anisomycin treatment can induce the cytosolic localization of cGAS in a ZNF598 regulated manner (Mol Cell. 2021 Jul 1;81(13):2808-2822.). This offers another readout of ribosome collision and ZNF598 function. In our previous studies we have effectively applied this system as an indicator of ribosome collision (Proc Natl Acad Sci U S A. 2022 Oct 18;119(42):e2202322119.).

Here we are using cGAS-STING signaling again as a readout of ribosome collision.

9) Similarly, the relevance of the stalled nascent chain in the nucleolus and then polysome fractionation is not at all clear.

The reviewer's point is well taken. The trafficking of stalled nascent chain from stalled ribosomes to the nucleolus is an integral part of the RQC of poly(GR) translation, but a complicated process accompanied by post-translational modifications such as CAT-tailing and possible proteolytic processing. We have now removed that data to avoid confusion.

10) What is FMRP?

FMRP is Fragile X messenger ribonucleoprotein 1 that is associated with polysomes and commonly used as a polysomal marker. We have mentioned its full name in the text and cited the reference on its polysomal localization.

11) The data in Figure 5C is not clear. It seems that ubiquitinated Rps3 is already visible in control cells, is this expected? Is there is so much ribosome collision that one sees this in control cells? Then the Flag signal in the gradient is very unconvincing and does not overlap with the fraction of most increased NOT4 signal.

The reviewer's point is well taken. Ubiquitinated RPS3 level should be low in control cells. The weak signals shown in control cells was likely due to long exposure of blots. We have repeated the experiment and presented improved images in Fig. 5d, e.

12) The authors use both NOT4 (text on figure 5) and CNOT4 (figure 5).

We thank the reviewer for pointing this out. We have now used CNOT4 throughout the manuscript.

13) The co-ip of NOT4 with Flag-GR80 is not super convincing since the negative control has nearly the same signal.

The reviewer's point is well taken. We have now repeated the experiment and provided improved images in Fig. 5b.

14) The major signal that is increased in Figure 6A seems to be a discrete species rather than the smear.

We understand the reviewer's point. From the previous data it might be difficult to clearly tell whether the smear band is ubiquitin or not. This is likely an antibody specificity issue. We therefore used a different setting in which we co-express FLAG-ZNF598 and HA-Ub in cells followed by rotenone and CCCP treatment. ZNF598 is enriched and Ub is detected using denaturing co-IP assay. Ubiquitinated ZNF598 is clearly shown as smear. The updated data is shown in Fig. 6a.

15) For the experiment to show that Flag-ZNF598 is ubiquitinated by HA-Ubi would be to do an HA-IP and reveal by Flag, rather than a Flag IP and reveal higher MW species by Flag (Figure 6E) and for Figure 6F, it

is the same, it should be HA IP revealed by Flag, because the smear could be co-purifying proteins that are ubiquitinated

The reviewer's point is well taken. We want to point out that in our experiments we performed IP experiment under denaturing conditions, so the concern of detecting associated proteins instead of the intended target protein is relieved. Nevertheless, we also performed the reverse IP as the reviewer suggested by performing HA IP first to pulldown all ubiquitinated proteins and then detect ZNF598. However, due to the sheer number of ubiquitinated proteins in cells and the limited sensitivity of the available ZNF598 antibody, it has been technically challenging to detect ZNF598 in the IP.

In Fig. 6e, we have replaced the original image with new images showing Flag IP followed by HA western blot of cells co-expressing HA-Ub with FLAG-ZNF598 or FLAG-ZNF598 C29A. The new image better demonstrates high MW Flag signals that are HA-Ub positive.

16) Figure 7A is not convincing, and it is overall not convincing that NOT4 levels are limiting and an effect can be seen by its overexpression. In Figure 7E, 7F, and 7H, the effect of overexpressing NOT4 seems very variable. It could be that overexpression of NOT4 itself is variable or that NOT4 works with other proteins (the CCR4-NOT complex for instance) and overexpressed alone will not have a major effect. Do the authors know if it works in isolation or not? Instead, the depletion (Figure 7B) is convincing and it seems that it is the discreet ubiquitin band rather than the smear that decreases. I would certainly put more emphasis of this.

The reviewer's point is well taken. Indeed, the overexpression effect of CNOT4 on translational readthrough in the reporter assays was variable. This is likely due to variable expression levels of CNOT4 achieved. We have now repeated these experiments by controlling the expression level of CNOT4 to make it more consistent. The effect on translational readthrough in the reporter assays is now less variable. This is shown in Fig. 7e, f, and h.

17) For Figure 7J, I do not see a synergy but a rather modest additive effect.

The reviewer's point is well taken. In the original data, too much ZNF598 expression alone might have offered strong protective effect in the stress assay, making the synergetic effect hard to appreciate. We have therefore repeated the experiment by overexpressing ZNF598 at a lower level. As shown in Fig. 7I, CNOT4 can synergistically promote the protective effect of ZNF598 under this condition.

18) In the discussion, the authors refer to yeast, but should mention "fission yeast".

Thanks for the suggestion. We have changed that.

19) Does NOT4 impact the stability of ZNF598 since it impacts its ubiquitination, or is it like for ABCE1, not impacting the stability?

Overexpression of CNOT4 doesn't impact the stability of ZNF598, as shown by the lack of significant change of ZNF598 protein level under CNOT4 overexpression condition as shown in Fig 7a. We have also discussed this point on page 17.

Reviewer #3 (Remarks to the Author):

Review for the manuscript NCOMMS-23-02506 by Dr. Bingwei Lu and co-authors entitled "A NOT4-ZNF598 axis responds to mitochondrial stress to quality control stalled translation and maintain tissue homeostasis".

Translational stalling events that result in ribosome collisions induce Ribosome-associated Quality Control (RQC) to degrade potentially toxic truncated nascent proteins. For RQC induction, the collided ribosomes are first marked by the ZNF598 E3 ubiquitin ligase to recruit the RQT complex for subunit dissociation. ZNF598 functionally marks collided mammalian ribosomes by K63-linked polyubiquitination of uS10 for the trimeric hRQT complex-mediated subunit dissociation. Although, the triggering steps of RQC are well-characterized, however, the physiological regulation of RQC in general and ZNF598 in particular in multicellular settings is underexplored.

In this study, the authors demonstrated the results suggesting that ZNF598 undergoes regulatory K63-linked ubiquitination under mitochondrial stress (Figure 6), and the NOT4-ZNF598 axis regulates quality control of stalled translation and stress response (Figure 7). Based on these, they proposed that ZNF598 undergoes regulatory K63-linked ubiquitination in a NOT4-dependent manner and is upregulated upon mitochondrial stresses in mammalian cells and *Drosophila*. The proposal is potentially interesting. However, the quality of the results is not sufficient enough to support their proposal. In special, the quantification of the signals is crucial in this manuscript. The signals of the binds detected by chemiluminescence/autoradiography must be quantified using the calibration curve. The main conclusion of this manuscript must be clarified by showing the phenotype of the ZNF598 mutant defective in NOT4-mediated ubiquitination.

We appreciate the reviewer's comment on our proposed regulatory K63-linked ubiquitination of ZNF598 in a NOT4-dependent manner in response to mitochondrial stresses in mammalian cells and *Drosophila* as potentially interesting. In the revised manuscript, we have quantified the immunoblot signals throughout the manuscript to further support our proposal. We have also made a ZNF598 mutant defective in CNOT4-mediated ubiquitination and tested its function in mammalian cells. Our new results support that NOT4-mediated ubiquitination of ZNF598 is critical for the quality control of stalled translation.

Comments:

1. The main conclusion in this manuscript depends on the precise quantification of the signal intensities on the western blots. The signals of the binds detected by chemiluminescence/autoradiography must be quantified using the calibration curve. It is mandatory to demonstrate the quantification of the signal intensities using the calibration curve in all figures.

The reviewer's comment is well taken. The signals on immunoblots depend on the exposure time, background noise, and loaded protein amount. One way to quantify the signal intensities, as suggested by this reviewer, is to generate a standard calibration curve for each experiment, such that the absolute abundance of each protein can be measured and compared with data from different experiments. In our experiments, we carefully controlled sample loading and exposure time so that we can generate immunoblots with protein amounts and signal intensities in the linear range. We then use NIH ImageJ to measure the relative expression when compared to control group or treatment groups in the same blot. This is commonly used in the literature and in our previous publications. In our experience with this method, results from

different experiments show the same trend and are consistent, leading us to draw firm conclusion from these data. We have described our method in more detail in Methods section on page 23-24. We hope the reviewer will agree that our approach is acceptable.

2. Based on the results shown in Figure 7, the authors propose that NOT4-dependent K63-linked ubiquitination of ZNF598 is required for resistance to mitochondrial stress. However, there is no direct evidence to demonstrate that Not4 directly ubiquitinates ZNF598, and the K63-linked polyubiquitination of ZNF598 somehow affects cell viability. The ubiquitination defective ZNF598 mutant must be constructed by the identification of lysine residue sites of NOT4-mediated ubiquitination. Moreover, the *in vitro* ubiquitination of ZNF598 by NOT4 may clearly show the linkage specificity.

The reviewer's point is well taken. On page 18, we present the following: *Based on the information that the N terminus of ZNF598 is critical for RQC²⁶, and that many of the lysine residues predicted by PhosphoSitePlus to be involved in ZNF598 ubiquitination are located at the N-terminus, we made the ZNF598 KR mutant with the predicted N-terminal ubiquitination sites mutated (Fig. S6f). The ubiquitination of ZNF598 KR mutant by CNOT4 was indeed decreased compared to ZNF598 WT (Fig. 7i), and it failed to block the readthrough of stalled ribosomes either alone or together with CNOT4 (Fig. 7j). These data support the functional significance of ZNF598 ubiquitination by CNOT4 in RQC. We also show that compare to ZNF598-WT, ZNF598-KR is less effective in protecting against mitochondrial stress in cell viability assays (Fig. S6g).*

3. In Figure 5, the authors show the co-IPed RQC factors are co-IPed with the FLAG-GR80. Genetic and biochemical research has been clarifying the substrate specificity of ZNF598-mediated ubiquitination and ASCC3/ASCC2/TRIP4-mediated splitting of the collided ribosomes. It is necessary to demonstrate that ZNF598 and ASCC3 are recruited to the ribosome stalled during the translation of FLAG-GR80 mRNA. The immunoprecipitated samples with FLAG-GR80 should be subjected to sucrose gradients to identify the disome or trisomes. The ribosome proteins should be detected with the CBB stain of the IPed samples. Moreover, FLAG-GR80 reduces mCherry by an unknown mechanism in Figure 5F. One of the required control experiments is the measurement of GFP and mCherry in ZNF598-KO.

The reviewer's point is well taken. We have tried the experiment suggested by the reviewer. However, to perform the suggested experiments, a sufficient amount of Flag-GR80 mRNA/ribonucleoprotein (mRNP) complex needs to be prepared and fractionated by sucrose gradient. The feasibility of this experiment is limited by Flag-GR80 mRNP preparation due to reagent availability. Instead, to test whether ZNF598 and ASCC3 are recruited to collided ribosomes in GR80 expressing cells, we used RNase A to digest ribosomes followed by sucrose gradient fractionation to enrich collided ribosomes. As shown in Fig. 5e, ZNF598 and ASCC3 were shifted to Lane 5, and Lane 4 and 5, respectively. The ub-RPS3 species were also enriched in lane 4 and 5 in GR80 group, supporting that these fractions contain collided ribosomes.

To test whether the reduced mCherry is due to GR80 induced stalling, ZNF598 KO cells are used and the GFP/mCherry ratio measured. As shown in Fig. 5c, ZNF598 KO can totally remove the inhibitory effect of GR80 on the readthrough of K20, supporting that Flag-GR80 reduces mCherry experiment by promoting ribosome stalling and reduced translation of stalled mRNA through ZNF598.

4. Figure 6H, the difference in the mCherry/GFP ratios between with or without HA-Ub K63 expression is relatively small (0.47, lane 2 vs 0.24, lane 4). It is mandatory to perform the control experiments with Ub-K63R mutants.

We thank the reviewer for the suggestion. We have added Ub-K63R as a control. The updated data is shown in Fig. 6h.

5. In Figures 7F-G, the ribosome stalling at K20 in NOT4-KO is attenuated but increased by over expression of Not4. The precise quantification of GFP and mCherry products is crucial, therefore the measurement of fluorescence intensities of GFP/mCherry with flow-cytometry is highly recommended.

We thank the reviewer for the constructive suggestion. We have now performed the flow-cytometry experiment. The fluorescence of GFP/mCherry as detected using flow-cytometry is now shown in Fig. S6d, e. The fluorescence intensity data obtained by flow-cytometry agrees with western blot data.

6. In Figure 2B, the level of mKate2 an indicator of ribosome stalling differs different in ZNF598-KO and ASCC3-KO cells. In principle, the ZNF598 and ASCC3 contribute to the splitting of the collided ribosomes, it's better to explain the reason why ribosome stalling is reduced differently in these KO cells.

The reviewer's point is well taken. There are two potential explanations for the differing effects of ZNF598 and ASCC3 inhibition on readthrough of the translation stall as indicated by mKate 2 level. 1) We knocked out ZNF598 using CRISPR-Cas9 but knocked down ASCC3 by RNAi. The different degrees of inhibition of ZNF598 and ASCC3 by knockout vs knockdown might contribute to the differential effects on translational readthrough. We also applied flow-cytometry to detect the GFP and mKate signal as show in Fig. 2c and Fig. S2d. The flow-cytometry data largely agrees with the Western blot dat; 2) As we explained in Response to Point 7 of Reviewer 1, *OE of a GFP-ZNF598 construct only partially rescued mCherry expression in ASCC3 deficient cells (Fig. 2d). On the other hand, OE of Flag-ASCC3 did not significantly affect mCherry expression in ZNF598 KO cells (Fig. 2e), suggesting that ZNF598 may act downstream of ASCC3, but ZNF598-independent function of ASCC3 or ASCC3-independent function of ZNF598 may also exist.*

7. In Figure 2A, the cell viability differs between Ctrl and ZNF598-KO without any treatment. Since the effects of 'the none' in cell viability by ZNF598-KO are significant in comparison with CCCP treatment, the p-value of the none must be demonstrated.

We thank the reviewer for the suggestion. We have added the p-value for the ctrl group ($p=0.0766$) in Fig. 2a.

Reviewer #4 (Remarks to the Author):

In this manuscript the authors investigate the role of the protein ZNF598 in translation stalling and mitochondrial stress. They use a combination of mammalian cell, *Drosophila* and mouse models combined with genetic manipulation of ZNF598 and other factors involved in ribosome-associated quality control. They find that ZNF598 expression is increased by mitochondrial stress, acting downstream of ASCC3. ZNF598 is localised to mitochondria in *Drosophila* neurons and overexpression of ZNF598 rescues the phenotypes in Pink1 mutant flies. ZNF598 overexpression also rescues the phenotypes in *Drosophila* models and patient cells expressing the G4C2 C9orf72 repeats. They also show that ZNF598 levels are regulated through ubiquitination induced by mitochondrial stress. This is an interesting study and reveals new mechanisms associated with mitochondrial stress and the link to translation stalling through ZNF598. The manuscript is well written and the figures are clearly presented.

We thank the reviewer for the overall positive comments on our manuscript. We reviewers' remaining comments are addressed below.

I have the following comments:

1. The authors propose that ZNF598 and the regulation of ribosome stalling in response to mitochondrial stress are localised to the mitochondrial outer membrane. Although the authors show that ZNF598 is localised to mitochondria in *Drosophila* neurons more evidence is required to support the proposed mechanism. The authors should analyse ZNF598 localisation in mammalian cells. They should also analyse whether ZNF598 is localised to the mitochondrial outer membrane, intermembrane space or mitochondrial matrix using, for example, cellular fractionation studies combined with protease treatment of isolated mitochondria.

We thank the reviewer for the constructive suggestion. We have now further analyzed the localization of ZNF598 in mammalian cells. We applied MPTP-induced mitochondrial damage model and isolated mitochondria to detect ZNF598 distribution in Fig. 1g. To further analyze ZNF598 localization in mammalian cells, co-staining of endogenous ZNF598 and a mitochondrial marker (TOM20) was performed in U2OS cells followed CCCP treatment or MTS GFP-K20 stall reporter expression (Fig. 3a). The immunofluorescent images show that ZNF598 is enriched in mitochondria after mitochondrial stress.

To detect the specific location of ZNF598 in mitochondria, mitochondria from control and CCCP treatment cells were isolated and incubated with Proteinase K with or without Triton X-100 presence. Mitochondrial associated ZNF598 was dramatically reduced by Proteinase K treatment in the absence of Triton X-100, suggesting that it is localized to mitochondrial outer membrane, consistent with its role in handling the quality control of mitochondrial surface localized stalled ribosomes upon mitochondrial stress (Fig. 3b).

2. Expression levels are quantified beginning in Figure 1 and throughout the manuscript without statistical analysis, e.g. Figure 1A-F. If these quantifications are representative of several experiments, then this should be stated where appropriate in the figures. Also, in the first Results paragraph it is claimed that CCCP causes a 'significant' increase in ZNF598 expression. This gives the impression that statistical analysis has been performed. The increase should either be shown to be statistically significant, or this word should be removed

here and elsewhere, e.g. p.7.

The reviewer's point is well taken. We have now presented statistical analysis of protein expression levels in Fig. 1 and Fig. 4f.

3. The GFP-P2A-Flag-K20-P2A-mKate2 reporter is used extensively but this was designed as a reporter of cytosolic translation. The authors suggest that at least some translation involving the reporter would occur on the mitochondrial outer membrane. Can this be expanded upon? Is it known what proportion of translation occurs on the mitochondrial outer membrane? It seems surprising that the reporter is so sensitive to changes in ZNF598 levels when this probably only regulates a small proportion of overall translation.

The reviewer's point is well taken. As shown in Fig. S2e, the MTS-GFP colocalizes with mitochondria, and intriguingly, a significant portion of mKate2 also colocalizes with mitochondria (Fig. 2h). Since mKate itself has not MTS, this result suggests translation of the mitochondrial-targeted stall reporter on mitochondrial surface and co-translational import of the reporter proteins. The MTS-GFP-K20 and mKate2 proteins localized to mitochondria were increased in ZNF598 KO cells as shown by immunostaining (Fig. 2h) or immunoblots of isolated mitochondria (Fig. 2i), supporting the role of ZNF598 in regulating mitochondrial outer membrane associated stalled translation. We would also like to point out that ZNF598 regulates both cytosolic and mitochondrial-associated stalled translation, as the effect of ZNF598 KO on the readthrough of the translational stall was comparable in the cytosolically localized GFP-P2A-Flag-K20-P2A-mKate2 reporter and the MTS-GFP-K20-P2A-mKate reporter systems based on the relative mCherry signal (Fig. 2f). Our data show that ZNF598 function in regulating mitochondrial outer membrane associated translation is regulated by both its recruitment to mitochondrial outer membrane and its posttranslational modification in response to stress.

4. On p.12 it is stated that the effect of overexpression of ZNF598 and other RQC factors on C-130-u formation is tested. However, only data for HA-Rack1 is shown in Figure 3B. Data for the other RQC factors should be included, or this statement modified.

We understand the reviewer's point. We have now added data for other RQC factors in C-130 u formation in Fig. S3b.

5. The experiment shown in Figure 4F should be analysed statistically with repeats.

We thank the reviewer for the suggestion. We have repeated the experiment and performed statistical analysis. The quantification data is shown in Fig. 4f.

6. It would be very helpful to include a model for the proposed mechanism of regulation of ribosome stalling by ZNF598 in response to mitochondrial stress in the figures.

We have now added a model as Fig. S7.

Minor comments:

1. On p.8 it states that ZNF598 KO cells are more sensitive to all the mitochondrial stressors tested but the

decrease in viability with BFA treatment is not significant, so this should be mentioned.

We thank the reviewer for pointing this out. We inadvertently missed the statistical analysis in the BFA group. The difference is significant in the BFA group ($p=0.0226$). We have now indicated this in Fig. 2a. In addition, ZNF598 overexpressing cells were also treated with rotenone, CCCP, and Antimycin A + Oligomycin, and cell viability data is shown in Fig. S2b.

2. A citation should be included for the GFP-P2A-Flag-K20-P2A-mKate2 reporter on p.9. Also, a more details description of this reporter would be helpful.

We thank the reviewer for the suggestion. We have now cited the reference for the GFP-P2A-Flag-K20-P2A-mKate2 reporter and added more detailed description of the reporter on page 8-9.

3. Scale bars should be added to all immunofluorescence images.

We thank the reviewer for the suggestion. We have now added scale bars in all IF images.

REVIEWER COMMENTS

Reviewer #1 (Remarks to the Author):

The revised manuscript by Geng et al. is very much improved and most issues have been addressed. I have only minimal comments remaining.

Minor comments

1) Line 144: this comment about higher up-regulation of RPS6 compared to ZNF598 is strange. It suggests that everything is up-regulated upon mitochondrial stress, and that the stoichiometry of ZNF598 to ribosomes explains why RPS6 is more up-regulated. Has an up-regulation of ribosomes and associated factors upon mitochondrial stress been described? If so, a citation is needed.

2) In the experiment in Figure 5d, FMRP seems to be overexpressed not just shifted in polysome fractions (this could be a consequence of overexpression). This needs to be commented on.

3) Line 400, 401: it should be written CNOT4-dependent, not CNOT4-mediated, as direct ubiquitination of ZNF598 by CNOT4 is not shown.

4) A role for Not4 in quality control at the mitochondrial outer membrane in yeast has recently published and it should be referenced, at least in the discussion (doi: 10.1093/nar/gkad299) because it appears to be very directly related to the mechanism described in this manuscript but dissected in budding yeast.

Reviewer #3 (Remarks to the Author):

Review for the manuscript NCOMMS-23-02506A by Dr. Bingwei Lu and co-authors entitled "A NOT4-ZNF598 axis responds to mitochondrial stress to quality control stalled translation and maintain tissue homeostasis".

The authors have addressed most of my comments; however, the critical point remains unresolved. The primary concern was that the revised version did not provide experimental evidence to support the main conclusions of the manuscript. Specifically, the phenotype of the ZNF598 mutant defective in NOT4-mediated ubiquitination, which is crucial to the main proposition of the manuscript, has not been demonstrated in the revised version. While the authors predicted Not4-mediated ubiquitination sites in ZNF598 in silico and showed a 57% reduction in ZNF598 ubiquitination for the ZNF598-KR mutant under NOT4 overproduction, the functionality of this mutant in the proposed CNOT4-ZNF598 axis remains unaddressed.

Overall, there are overstatements and biochemical evidence is missing to support the proposal that Not4 ubiquitinates ZNF598 and modifies RQC. I recommend the review of this MS by the expert of RQC/ribosome collision and/or ubiquitination to maintain the quality of the papers published in Nature Communications.

Comments:

1. To substantiate their main proposal, as indicated in the title "A CNOT4-ZNF598 axis responds to mitochondrial stress to quality control stalled translation and maintain tissue homeostasis," it is essential to investigate quality control of stalled translation and stress response in cells expressing ZNF598-WT or ZNF598-KR, under ZNF598-KO or KD condition. In Figure 7j, the authors monitored ribosome stalling during the overproduction of ZNF598-WT or -KR in combination with NOT4 overproduction. Therefore, it is possible to explore the ribosome stalling phenotype of the ZNF598 mutant defective in NOT4-mediated ubiquitination using siRNA-resistant ZNF598-WT or ZNF598-KR. In Figure 7k, the authors monitored cell viability under the ZNF598 siRNA condition in combination with NOT4 overproduction. Therefore, it is possible to explore the cell viability phenotype of the ZNF598 mutant defective in NOT4-mediated ubiquitination using siRNA-resistant ZNF598-WT or ZNF598-KR.

2. The subtitle in line 342, "CNOT4-dependent K63-linked ubiquitination of ZNF598 upon mitochondrial stress," is not supported by the text in lines 343-367. There is no mention of NOT4-related results or of the word "NOT4" in lines 343-367.

3. There is an overstatement in line 371: "Given our data indicating CNOT4 involvement in handling stalled translations (Fig. 5)." The only suggestive result is the possible distribution of CNOT4 in the disome, as shown in Fig 5e. Under RNase-treated conditions, ZNF598 was distributed in fraction 5 in FLAG-GR80-transfected cells but not in control cells, suggesting that the RNase-resistant disome

may be in fraction 5. However, it should be validated whether the disome is distributed in Fraction 5. The authors should provide the A254 absorbance data.

4. In line 465, I would like to clarify that the yeast utilized in references 69 and 70 is not fission yeast, but rather budding yeast, *S. cerevisiae*. Additionally, in reference 68, the authors provided a comprehensive summary of the findings related to *S. cerevisiae*.

5. There were significant disparities between the original and revised Figure 7c (WB with ZNF598 antibody) and 7h (WB with mCherry antibody). In Figure 7c, although the three blots remain unaltered, the initially detected 150kDa ZNF598 bands in the original figure show a substantial reduction. In the revised Figure 7h, while the three blots remained consistent, the levels of mCherry bands under NOT4 overproduction conditions were notably diminished. However, the authors should clarify these inconsistencies.

Figure 7c (left panels, the original; the right panels, the revised)

Figure 7h (left panels, the original; the right panels, the revised)

Reviewer #4 (Remarks to the Author):

I am happy with the responses to my previous comments and the revised manuscript.

Point-by-Point Response to Reviewer Comments on NCOMMS-23-02506A

Reviewer #1 (Remarks to the Author):

The revised manuscript by Geng et al. is very much improved and most issues have been addressed. I have only minimal comments remaining.

Minor comments

1) Line 144: this comment about higher up-regulation of RPS6 compared to ZNF598 is strange. It suggests that everything is up-regulated upon mitochondrial stress, and that the stoichiometry of ZNF598 to ribosomes explains why RPS6 is more up-regulated. Has an up-regulation of ribosomes and associated factors upon mitochondrial stress been described? If so, a citation is needed.

The reviewer's point is well taken. We have now referenced our previous publications showing up-regulation of ribosomes and associated factors upon mitochondrial stress on page 7.

2) In the experiment in Figure 5d, FMRP seems to be overexpressed not just shifted in polysome fractions (this could be a consequence of overexpression). This needs to be commented on.

We thank the reviewer for pointing this out. Indeed, in our other unpublished work, we found that FMRP is slightly increased in GR80 overexpressing cells. To avoid the confusion of this overexpression effect, we have calculated the proportion of FMRP in each lane relative to the whole amount across the polysome fractions in control and GR80 overexpressing cells. We have commented on this on page 15.

3) Line 400, 401: it should be written CNOT4-dependent, not CNOT4-mediated, as direct ubiquitination of ZNF598 by CNOT4 is not shown.

The reviewer's point is well taken. We have now revised it to 'CNOT4-dependent'.

4) A role for Not4 in quality control at the mitochondrial outer membrane in yeast has recently published and it should be referenced, at least in the discussion (doi: 10.1093/nar/gkad299)IF: 14.9 Q1 because it appears to be very directly related to the mechanism described in this manuscript but dissected in budding yeast.

We thank the reviewer for bring to our attention this study in yeast. It is indeed relevant and we have cited it in our Discussion on page 22.

Reviewer #3 (Remarks to the Author):

Review for the manuscript NCOMMS-23-02506A by Dr. Bingwei Lu and co-authors entitled "A NOT4-ZNF598 axis responds to mitochondrial stress to quality control stalled translation and maintain tissue homeostasis".

The authors have addressed most of my comments; however, the critical point remains unresolved. The primary concern was that the revised version did not provide experimental evidence to support the main conclusions of the manuscript. Specifically, the phenotype of the ZNF598 mutant defective in NOT4-mediated ubiquitination, which is crucial to the main proposition of the manuscript, has not been demonstrated in the revised version. While the authors predicted Not4-mediated ubiquitination sites in ZNF598 in silico and showed a 57% reduction in ZNF598 ubiquitination for the ZNF598-KR mutant under NOT4 overproduction, the functionality of this mutant in the proposed CNOT4-ZNF598 axis remains unaddressed.

Comments:

1. To substantiate their main proposal, as indicated in the title "A CNOT4-ZNF598 axis responds to mitochondrial stress to quality control stalled translation and maintain tissue homeostasis," it is essential to investigate quality control of stalled translation and stress response in cells expressing ZNF598-WT or ZNF598-KR, under ZNF598-KO or KD condition.

In Figure 7j, the authors monitored ribosome stalling during the overproduction of ZNF598-WT or -KR in combination with NOT4 overproduction. Therefore, it is possible to explore the ribosome stalling phenotype of the ZNF598 mutant defective in NOT4-mediated ubiquitination using siRNA-resistant ZNF598-WT or ZNF598-KR.

We thank the reviewer for this constructive suggestion. The ZNF598 shRNA targeting the 3'UTR of endogenous ZNF598, which does not affect the ZNF598 WT and ZNF598 KR constructs, was applied in our experiments. As shown in new data added as **Figure 7m**, while ZNF598-WT overexpression could decrease the readthrough of K20 stalling reporter in CNOT4 overexpressing cells with endogenous ZNF598 knocked down, ZNF598 KR had no effect. This result suggests that the CNOT4-ZNF598 axis relies on the ubiquitination of ZNF598 to remove stalled ribosomes.

In Figure 7k, the authors monitored cell viability under the ZNF598 siRNA condition in combination with NOT4 overproduction. Therefore, it is possible to explore the cell viability phenotype of the ZNF598 mutant defective in NOT4-mediated ubiquitination using siRNA-resistant ZNF598-WT or ZNF598-KR.

We also applied the 3'UTR targeting ZNF598 shRNA in combination with ZNF598 WT and ZNF598 KR in cell viability assays with CCCP treatment. As shown in the new data added as **Figure 7n**, knocking down endogenous ZNF598 blocked the CNOT4 effect on cell viability. Introduction of ZNF598 WT, but not ZNF598 KR, promoted

cell viability against CCCP in CNOT4 overexpressing cells with endogenous ZNF598 knocked down, suggesting that CNOT4 depends on ZNF598 and ZNF598 ubiquitination to protect cells against mitochondrial stress.

2. The subtitle in line 342, "CNOT4-dependent K63-linked ubiquitination of ZNF598 upon mitochondrial stress," is not supported by the text in lines 343-367. There is no mention of NOT4-related results or of the word "NOT4" in lines 343-367.

The reviewer's point is well taken. We have removed "CNOT4-dependent" from the subtitle. The subtitle now becomes "K63-linked ubiquitination of ZNF598 upon mitochondrial stress".

3. There is an overstatement in line 371: "Given our data indicating CNOT4 involvement in handling stalled translations (Fig. 5)." The only suggestive result is the possible distribution of CNOT4 in the disome, as shown in Fig 5e. Under RNase-treated conditions, ZNF598 was distributed in fraction 5 in FLAG-GR80-transfected cells but not in control cells, suggesting that the RNase-resistant disome may be in fraction 5. However, it should be validated whether the disome is distributed in Fraction 5. The authors should provide the A254 absorbance data.

The reviewer's points are well taken. We have now changed "indicating" to "suggesting" in the sentence in question on page 17. In our previous data, we used the distribution of ZNF598 and Ub-RPS3 to deduce the position of collided ribosomes. Now, we further used Piston Gradient Fractionator to detect the disomes by A254 absorbance as suggested by the reviewer. The absorbance curve is now integrated into **Fig. 5e**.

4. In line 465, I would like to clarify that the yeast utilized in references 69 and 70 is not fission yeast, but rather budding yeast, *S. cerevisiae*. Additionally, in reference 68, the authors provided a comprehensive summary of the findings related to *S. cerevisiae*.

We thank the reviewer for pointing this out. We have now changed the species from fission yeast to **budding yeast**.

5. There were significant disparities between the original and revised Figure 7c (WB with ZNF598 antibody) and 7h (WB with mCherry antibody). In Figure 7c, although the three blots remain unaltered, the initially detected 150kDa ZNF598 bands in the original figure show a substantial reduction.

We thank the reviewer for pointing this out. Two antibodies raised against ZNF598 are used in our work. One is the ZNF598 antibody [N1N3] from GeneTex, the other is ZNF598 antibody HPA041760 from Sigma Aldrich. Both antibodies can detect ZNF598 bands around 100 kD and 150 kD, but for reasons unclear to us, they show different affinities to these bands. To maintain consistency, we replaced the previous blots detected with N1N3 from GeneTex with the one detected with HPA041760 from

Sigma Aldrich.

In the revised Figure 7h, while the three blots remained consistent, the levels of mCherry bands under NOT4 overproduction conditions were notably diminished. However, the authors should clarify these inconsistencies.

The reviewer's point is well taken. Apparently, the efficiency of CNOT4 overexpression is critical for experimental outcome when multiple plasmids are co-transfected into cells. The slight reduction of mCherry shown in previous data was most likely due to inefficient expression of CNOT4 in the multi-plasmid transfection experiment, although we did not specifically detect CNOT4 expression at that time. When we repeated the experiment, we adjusted the amounts of plasmids used and we detected NOT4 expression to confirm that CNOT4 overexpression was achieved. The new data is more consistent as judged by CNOT4 expression and we believe that it reflects the real situation.

REVIEWERS' COMMENTS

Reviewer #1 (Remarks to the Author):

In this revised version the authors have answered all the remaining concerns that I had and they also improved their manuscript by addressing the concerns of another reviewer.

My opinion is that this manuscript provides interesting new data that together with only two other manuscripts that I am aware of, discusses ribosome quality control regulation at the mitochondrion, and additionally in this manuscript now they place this in physiological context. I feel that this will be of broad interest for the readers in many different fields.

Reviewer #3 (Remarks to the Author):

Review for the manuscript NCOMMS-23-02506B entitled "A NOT4-ZNF598 axis responds to mitochondrial stress to quality control stalled translation and maintain tissue homeostasis"

The authors have addressed most of my previous concerns. I support the publication of manuscript in Nature Communications.